# The Hitchhiker’s Guide to Human Therapeutic Nanoparticle Development

**DOI:** 10.3390/pharmaceutics14020247

**Published:** 2022-01-21

**Authors:** Thelvia I. Ramos, Carlos A. Villacis-Aguirre, Katherine V. López-Aguilar, Leandro Santiago Padilla, Claudia Altamirano, Jorge R. Toledo, Nelson Santiago Vispo

**Affiliations:** 1Laboratorio de Biotecnología y Biofármacos, Departamento de Fisiopatología, Facultad de Ciencias Biológicas, Universidad de Concepción, Víctor Lamas 1290, Concepción 4070386, Chile; tiramos@espe.edu.ec (T.I.R.); cvillagui@outlook.es (C.A.V.-A.); 2Grupo de Investigación en Sanidad Animal y Humana (GISAH), Carrera Ingeniería en Biotecnología, Departamento de Ciencias de la Vida y la Agricultura, Universidad de las Fuerzas Armadas—ESPE, Sangolquí 171103, Ecuador; 3Carrera Ingeniería en Biotecnología, Departamento de Ciencias de la Vida y la Agricultura, Universidad de las Fuerzas Armadas—ESPE, Sangolquí 171103, Ecuador; kvlpeza@espe.edu.ec; 4Faculty of Biological Sciences, Friedrich-Schiller-Universität, 07743 Jena, Germany; saintpad97@gmail.com; 5Escuela de Ingeniería Bioquímica, Facultad de Ingeniería, Pontificia Universidad Católica de Valparaíso, Av. Brasil 2085, Valparaíso 2362803, Chile; claudia.altamirano@pucv.cl; 6Centro Regional de Estudios en Alimentos Saludables, Av. Universidad 330, Placilla, Sector Curauma, Valparaíso 2340000, Chile; 7School of Biological Sciences and Engineering, Yachay Tech University, Hda. San José s/n y Proyecto Yachay, Urcuquí 100119, Ecuador

**Keywords:** nanoparticles, nanomedicine, regulatory aspects, pharmacokinetics, preclinical, immunotoxicity, clinical trials

## Abstract

Nanomedicine plays an essential role in developing new therapies through novel drug delivery systems, diagnostic and imaging systems, vaccine development, antibacterial tools, and high-throughput screening. One of the most promising drug delivery systems are nanoparticles, which can be designed with various compositions, sizes, shapes, and surface modifications. These nanosystems have improved therapeutic profiles, increased bioavailability, and reduced the toxicity of the product they carry. However, the clinical translation of nanomedicines requires a thorough understanding of their properties to avoid problems with the most questioned aspect of nanosystems: safety. The particular physicochemical properties of nano-drugs lead to the need for additional safety, quality, and efficacy testing. Consequently, challenges arise during the physicochemical characterization, the production process, in vitro characterization, in vivo characterization, and the clinical stages of development of these biopharmaceuticals. The lack of a specific regulatory framework for nanoformulations has caused significant gaps in the requirements needed to be successful during their approval, especially with tests that demonstrate their safety and efficacy. Researchers face many difficulties in establishing evidence to extrapolate results from one level of development to another, for example, from an in vitro demonstration phase to an in vivo demonstration phase. Additional guidance is required to cover the particularities of this type of product, as some challenges in the regulatory framework do not allow for an accurate assessment of NPs with sufficient evidence of clinical success. This work aims to identify current regulatory issues during the implementation of nanoparticle assays and describe the major challenges that researchers have faced when exposing a new formulation. We further reflect on the current regulatory standards required for the approval of these biopharmaceuticals and the requirements demanded by the regulatory agencies. Our work will provide helpful information to improve the success of nanomedicines by compiling the challenges described in the literature that support the development of this novel encapsulation system. We propose a step-by-step approach through the different stages of the development of nanoformulations, from their design to the clinical stage, exemplifying the different challenges and the measures taken by the regulatory agencies to respond to these challenges.

## 1. Nanoparticles in Medicine

The design and fabrication of nanoparticles (NPs) as drug delivery systems is a promising and rapidly developing area which aims to provide and maintain therapeutic drug concentrations at the site of biological interaction [1]. These formulation designs improve aqueous solubility, increase chemical stability, and increase the therapeutic index of pharmacological agents [2]. Their physicochemical properties allow sustained release, improved bioavailability, modified pharmacokinetics, and reduced side effects [3]. NPs have a larger contact surface area which increases their stability and reactivity. Moreover, their magnetic, electrical, and biological properties can be modified to obtain different sizes, shapes, compositions, and chemical characteristics on their surface [4]. From the biological point of view, particle size reduction has provided numerous benefits in drug delivery because many of the internal mechanisms of a cell occur naturally at the nanometer scale (10^−9^ m). Among the advantages that stand out are increased bioavailability of therapeutic drugs, improved dose–response curve, and safety relative to traditional drugs [5].

The development of nanotechnology products offers innovative therapeutic and diagnostic opportunities to meet medical needs. Nanoformulations increase the commercial value of medical devices, represent a robust tool for applying for personalized medicine, and can be instrumental in treating orphan drugs [6]. These systems entered the pharmaceutical industry due to the expiration of many blockbuster generic drug patents, coupled with the excessive cost of new drug discovery [7]. There is a great diversity of NPs, such as dendrimers, liposomes, micelles, nanocapsules, nanospheres, inorganic NPs, polymeric NPs, among others [8]. Dendrimers are branched polymers with unique topological and structural characteristics, as they have three parts: a focal core, building blocks with several inner layers with repeating units, and multiple peripheral functional groups [9]. Liposomes are spherical structures consisting of one or more lipid bilayers enclosing aqueous spaces [10]. Polymeric micelles are nanostructures formed by the spontaneous arrangement in an aqueous medium of amphipathic polymer macromolecules [11]. Micrometer and nanometer-scale encapsulations are transport systems that create a physical barrier to protect the active ingredient from the external environment [12]. Microparticle formulations increase the bioavailability of the drug but have several drawbacks: low encapsulation efficiency, abrupt or incomplete release of the active ingredient, reduced biological activity, among others [13]. Encapsulation in NPs consists of trapping active ingredients using a surrounding material [12]. This technology allows nanospheres (deposition systems that incorporate the active ingredient in the particle’s matrix) or nanocapsules (matrix consisting of the drug as the core and the particle material as the capsule) to be obtained [14]. Nanoencapsulation has been considered a more efficient delivery system than microscale systems, with better functionality because it exhibits greater drug protection, increased stability, higher loading capacity, superior encapsulation efficiency, sustained release, and improved bioavailability [15,16]. The limiting aspects of therapeutic use for nanoformulations are the control of drug release, opsonization of the particle, and the toxicity and immunogenicity they may cause [17,18]. In studying the relationship between polylactic acid-polyethylene glycol (PLA-PEG) particle size and its transport efficiency across the nasal mucosa, tetanus toxoid was encapsulated in particles of different sizes (200 nm, 1.5 μm, 5 μm, and 10 μm). The nasal bioavailability of tetanus toxoid encapsulated in 200 nm nanoparticles was higher than in larger particles [19].

Several elements are used to synthesize these structures, including proteins, peptides, polysaccharides, synthetic polymers, and various inorganic materials such as metals (gold, silver, iron, silicon) [20,21]. The encapsulating matrix characteristics influence the nanoformulation properties and depend on the material from which the particle was made [22]. Biocompatible and biodegradable materials are the first to consider when searching for a formulation that does not generate side effects. The great possibility of combinations provides a variety of conformations that can be easily altered by slight modifications, both in the raw materials and in the manufacturing processes [23].

Every product designed for medical use undergoes strict evaluation to demonstrate its safety and therapeutic efficacy; based on the regulatory approaches of national and international agencies, nanomedicines are no exception [24]. Nanoformulations are an area of innovation that has developed faster than regulatory frameworks [25]. These systems are regulated by existing regulatory frameworks for drugs and medical devices, but there is no specific regulatory structure [26]. Despite all the above advantages, few encapsulant systems are ever approved by regulatory agencies for therapeutic use, as they are very complex products that may raise regulatory issues regarding manufacturing quality, safety, and efficacy [27]. The most commonly described drawbacks are related to the delivery of the active ingredient due to the complex systemic administration [28]. Global regulatory trends for nanomedicines lack essential data on the manufacturing process, pharmacokinetics, pharmacodynamics, and immunotoxicity that demonstrates the product’s safety [29]. This lack of regulatory harmonization specific to nanoparticles has delayed their clinical use, difficulties that have been reported on numerous occasions [30,31]. Although the properties of NPs are transforming medical research [32], especially since the first drugs encapsulated in NPs were approved [33], many nanoformulations fail to achieve success in preclinical trials.

Consequently, there are few trials in clinical research [34] facing numerous regulatory challenges [35]. The question arises as to why nanoparticle-based biopharmaceutical systems fail to achieve marketing approval despite their significant advantages. The challenge lies in the fact that the very properties that make NPs promising have become a challenge for the researchers doing the design and the evaluators at regulatory agencies [32].

There are currently 58 nanoparticle therapies and imaging agents approved for clinical use by major regulatory agencies [36]. These formulations offer promising results for treating a wide variety of diseases such as cancer, infections, autoimmune disorders, cardiovascular, pulmonary, neurodegenerative, ocular (glaucoma) and regenerative therapy, among other applications. The most successful formulations to date have been polymers and lipids. The main advantages of nanoparticles are (1) an increased bioavailability due to improved water solubility, (2) increased resistance time in the body (increased half-life for clearance/increased specificity for their cognate receptors), and (3) the targeting of the drug to a specific region of the body (its site of action) [37].

Of the drugs approved by the major regulatory agencies, the United States Food and Drug Administration (FDA) and the European Medicines Agency (EMA), all use liposomal nanoparticle systems except for Abraxane, an albumin-bound paclitaxel nanoparticle [33]. In the case of proliferative or cancerous conditions, many approved formulations have been applied at various stages of the disease. Among these drugs, the following stand out: Doxil, liposomal doxorubicin functionalized with polyethylene glycol (PEG), liposomal daunorubicin (DaunoXo-me), liposomal vincristine (Marqibo) and liposomal irinotecan (Onivyde), liposomal doxorubicin without PEG (Myocet), and liposomal mifamurtide (MEPACT) [38]. Most of these formulations are non-PEGylated, despite the recognized advantages that this encapsulation system offers [39]. Marqibo^®^ (vincristine sulfate) is a liposomal formulation from sphingomyelin and cholesterol, significantly improving circulation time and accelerating dose escalation compared to standard Vincristine. The FDA approved this system to treat acute lymphocytic leukemia in adults [40]. Kadcyla^®^ (Herceptin^®^) is an antibody-drug conjugate for treating human epidermal growth factor receptor 2 (HER2+) breast cancer. The drug is delivered to cancer cells through recognition of the HER2 receptor (transtuzumab), and maytansine (DM1) triggers apoptosis [41].

Polymeric and non-polymeric nanoformulations and liposomes have also been developed for infectious diseases. Examples of these already approved drugs are Lipoquin™ (ciprofloxacin), Ambisome^®^ (amphotericin B), and others for lipid-containing amphotericin B, such as Abelcet and Visudyne [33,42]. Recent examples of liposomal vaccines against the coronavirus disease 2019 (COVID-19) are the Pfizer/BioNTech and Moderna COVID-19 vaccines, both of which are formed in liposomal nanoparticles (LNP) or PEGLips (artificial phospholipid vesicles effective for stabilizing pharmaceuticals). These vaccines make it possible to stabilize messenger RNA (mRNA) thanks to their lability [43,44].

In iron replacement therapeutics, nanoparticles have also significantly impacted iron concentrations in the body and are considered complex non-biological drugs [33]. In addition, nanoparticle systems in autoimmune conditions are promising as they target the inflamed tissue. Certolizumab pegol (CZP) is an example of a tumor necrosis factor-alpha (TNF-α) inhibitor widely used in the clinic with a half-life of 14 days [45].

We can conclude that in terms of therapeutic applications for nanoparticles, in 2016, the number of approved nanoparticles used in the clinic was 51 nanomedicines, and in another publication in November 2021, it appears updated to 58 [36].

Nanoformulations must demonstrate regulatory requirements created to evaluate drugs and could cause difficulties in their development [46]. They can pose approval challenges [47], and therefore their designers must consider parameters including physicochemical characterization, efficacy, pharmacology, toxicology, immunology, and hematology [32,48]. Challenges in terms of their physicochemical properties include structural attributes such as estimation of particle size, charge, composition, surface coatings, and determination of density, solubility, architecture, and surface ligand stability [29]. Its potentiality as a pharmaceutical product encompasses other analyses such as drug loading, pharmacological stability, drug release and conjugation, selection of appropriate in vitro and in vivo models, quantification, and assessment of biological activity [49]. For example, stability, which is necessary for all drugs, can be particularly complex in the case of nanomaterials (NMs), as several physicochemical processes, such as aggregation, agglomeration or separation, degradation, escape or release, etc., have to be taken into account [29,32,33,34,46,47,48,50].

The particularities of nanoparticles in medicine are also evident in the manufacturing, scaling, and quality control of these because there is no specific guidance in the choice of process control parameters and the selection of Critical Quality Attributes (CQAs) [27,51]. One of the major difficulties is toxicological studies. It is required to distinguish acute and chronic short- and long-term toxicity, mechanisms of damage and address other bioanalytical challenges related to the estimation of the free and particle-bound drug in biological matrices [52]. Immunotoxicity is one of the most explored areas in these formulations to gain insight into the properties of the carrier element that may lead to an exaggerated drug response [53]. There are no regulatory regulations for lifting this evidence to help the success of nanoformulations and to answer the question: What changes in its composition does the NP undergo when it interacts with biological fluids? [54]. The demonstration of their sterility and endotoxin levels should be presented differently for nanomedicines, considering their achievements in terms of dose reduction and frequency of administration [55]. Another relevant aspect is the choice of animal models, which should be of a species with physiological similarity to the actual intended system. However, many of the sensitive animal models for nanoparticles have ethical restrictions for their use [52]. Once NPs have reached clinical trials, other questions arise regarding the extrapolation of data obtained in preclinical trials due to physiological differences, dose estimation in humans, clinical trial design in terms of the number of sample size for a study with NPs, and how the treatment regimen will be developed in patients [56,57]. There are no standardized protocols for adapting physicochemical characterization, manufacturing process, biological activity design, or preclinical and clinical trials, making it difficult to establish the risk–benefit ratio required for any therapeutic product [58].

Chemically and biologically synthesized pharmaceuticals’ regulatory framework cannot govern nanoformulations [7]. These delivery systems have problems with clinical translation due to the lack of guidelines for assay development and the absence of specific regulatory aspects. There is a lack of controls, comparators, problems with stability, dose calculation, bioequivalence assessment, and biological toxicity demonstration [52]. The failure of several formulations at the clinical stage is due to the lack of specific protocols for physicochemical, biological, and physiological characterization [59]. There is a need to reconsider obtaining a broader data set to address the specificities of the pharmacokinetic and pharmacodynamic profiles of these novel formulations [60].

This manuscript describes the major challenges researchers have faced when creating a new formulation. Our work aims to provide helpful information to improve the success of nanomedicines by compiling the challenges described in the literature that support the development of this novel encapsulation system. Additional guidance is required to cover the particularities of this type of product, as some challenges in the regulatory framework do not allow an accurate assessment of NPs with sufficient evidence for clinical success. We also reflect on the current regulatory standards required to approve these biopharmaceuticals and the requirements demanded by regulatory agencies. We propose a step-by-step approach to the different stages of nanoformulation development, from initial design to the clinical stage, exemplifying the distinct challenges and the measures taken by regulatory agencies to respond to these challenges.

## 2. First Approaches of Regulatory Agencies in Nanoparticle Development

Efforts to overcome the challenges in the development of nanomedicines have led academia, industry, and regulatory agencies to maintain an open dialogue through forums, seminars, and talks [48]. In 2010, the first international scientific workshop on nanomedicines was held, with the participation of 27 countries [61]. Since then, the symposia have continued to review existing and emerging nanomedicines, analyzing aspects such as characterization, biodistribution, and interaction with biological systems to prepare for evaluating these products in the future and identifying development parameters with gaps in knowledge [62].

Agencies such as the FDA and EMA have created laboratories to standardize assays and provide advisory services in the design of nanomedicines, such as the Nanotechnology Characterization Laboratory (NCL) in the United States and the European Nanomedicine Characterization Laboratory (EU-NCL). These laboratories work to accelerate the development of nanoformulations with analytical assays to validate the safety of nano-drugs under regulatory standards [63]. A project called Regulatory Science Framework for Nano(bio)material-based Medical Products and Devices’ (REFINE), created in 2017, resulted in the publication of a White Paper summarizing the main challenges associated with the regulation of nanomedicines to guide research projects and the communities involved to advance in the regulatory field [64].

Different agencies, such as the EMA [65,66,67,68,69] and the Japanese Ministry of Health, Labor and Welfare (MHLW)/ Pharmaceuticals and Medical Devices Agency (PMDA) [70,71,72], have published several guidelines for nanotechnology-based medical products over the past few years, including Reflection Papers for liposomes, micelles, and nanoparticles (Table 1). The FDA (2017) provided a guidance document for the nanotechnology-based products industry [73]. To date, regulatory agencies recommend a case-by-case analysis, introducing specific trial modifications for each and using the same regulatory process as applied for conventional drugs [74].

## 3. Challenges in the Design and Physicochemical Characterization of Nanoformulations

Developing a new product with therapeutic potential is a logical and orderly process that gathers information, starting with research on the physicochemical properties and its biological activity, all the way to “proof of concept”, avoiding waste of resources and time [58]. There are several essential aspects to consider when presenting scientific evidence. Initial work should focus on the physicochemical characterization of the nanoformulation and the stability of its production, in addition to regulatory challenges and agency guidelines [51]. Biological activity demonstration during in vitro assays, i.e., cell lines, should correlate with the in vivo assays and selecting the proper tissue and animal model, considering the first aspects for the therapeutic product [65]. The demonstration of the different systems that regulatory agencies have approved has outlined the various challenges that have arisen [59]. In the following, we describe each of the parameters and their challenges in obtaining NPs for biomedical applications.

### 3.1. Physicochemical Characterization

Unlike the development of other therapeutic products, the evaluation of the toxicity potential of NPs in biological systems begins with the complete physicochemical characterization, being a critical step in the early stages of development, which contributes to the principles of quality, safety, and efficacy [73,76]. To make a product effective in the clinical setting, it is necessary to employ appropriate characterization technologies that correlate effect and biological consequences and predict toxicologic and therapeutic outcomes at the early stage of product development [77]. NPs are distributed in any organ or system across epithelial and endothelial barriers [78,79] and even reach the interior of cells by various mechanisms, diffusing them into cell membranes [80]. The toxicity of NPs is related to the evaluation of their physical and chemical characteristics [80] and their relationship with adverse events such as thrombosis and platelet aggregation (C60 fullerene (C60CS), single-wall nanotubes (SWNT), and multiple wall nanotubes (MWNT), carbon NPs used in drug discovery and delivery affect vascular hemostasis and precipitate causing thrombosis) [81]. Inflammation and neurodegenerative or circulatory disorders, among others, are also adverse events described [82]. They can enter cellular organelles (mitochondria, nucleus), altering metabolism and causing deoxyribonucleic acid (DNA) damage and cell death [83].

Functionalization is a strategy for modifying the physicochemical properties of NPs, which consists of the conjugation of molecules to the surface of the particles, such as folic acid, biotin, oligonucleotides, peptides, monoclonal antibodies, functional groups, among others [84]. This modification allows high precision to incorporate or alter specific properties in the NPs [85]. The functionalized particles possess non-invasive characteristics, anti-agglomeration, and good physical properties [84]. Binding can be performed through covalent and non-covalent bonds. With non-covalent conjugation, it is possible to make changes without affecting the structure of the mo-molecules. For example, Yue et al. (2019) developed a noncovalent functionalization process of curcubit-7-uril to gold NPs, which enabled image-driven chemo-photothermal therapy [86]. In the case of covalent bonding, structures with multiple functions aimed at diagnostic and therapeutic therapeutics can be obtained using linker molecules such as PEG [87,88]. An example is a study from Chen and co-workers (2017) who synthesized antigen-binding fragment (Fab)-conjugated micelles of antibodies, using PEG as a spacer to obtain a high Fab density on the surface of the NPs and achieve higher bioactivity [89].

By functionalization, it is possible to modify properties such as surface chemistry, hydrophobicity, or charge of the NPs to improve their solubility, biocompatibility, biodistribution, and clearance [90]. The two difficulties of NPs that have been mostly worked on with functionalization are uptake and biocompatibility, the efficiency of their uptake, the cytotoxic effects of the particles on cells, and their ability to cross biological barriers limit the clinical use of these systems [91]. The cellular uptake (active or passive) of NPs depends on their physicochemical properties. It is possible to increase the stability and reduce the aggregation of the particles through functionalizations with PEG, peptides, or zwitterionic ligands, to increase passive uptake. To achieve active and targeted cellular uptake, it can also be conjugated with antibodies, aptamers, carbohydrates, and proteins [85]. Fathian kolahkaj et al. (2019) reported effective uptake of poly lactic-co-glycolic acid (PLGA) NPs modified with monoclonal antibodies against HER2, with increased levels of internalization in HER2-positive cell lines [92]. Other ligands such as transferrin, insulin, and lipoproteins can bind to the surface of NPs to cross biological barriers such as the blood–brain barrier [93]. Lactoferrin functionalization of trimethylated PLGA-chitosan Nps encapsulating huperzine A had increased cellular uptake, with improved brain pharma transport [94]. In the case of biocompatibility, conjugation of molecules on the surface of NPs has been shown to improve formulations by modifying the surface charge and inactivation of chemical groups that destabilize the cell membrane [85]. For example, the addition of molecules such as human albumin allows the toxicity of the particles to be reduced. In 2020, Sanità et al. reported that albumin functionalization on silver-eumelanin hybrid NPs suppressed in vitro hemotoxicity in normal mammary cells (MCF10a) [95].

The main reason why their approval is limited for use in human therapeutics is due to incomplete particle analysis [96]. The characterization should include determining size, shape, composition, charge and surface chemistry, encapsulation efficiency, and evaluating the encapsulated drug (Figure 1) with parameters such as loading, distribution, release kinetics, interaction with cells, and transport system [97]. There is a general lack of specific protocols for the characterization of nanomedicines at physicochemical and biological levels [59], and guidelines are needed to assess the quality and safety of these emerging products [98]. Each of the NPs may require new or modified methods for their evaluation [48]. It is desirable to know each method’s advantages and disadvantages that provide reliable information for a specific parameter to determine the most appropriate selection [63,99].

Problems when choosing the most appropriate technique to establish the physicochemical properties appear from the beginning of the characterization. The most widely used method is dynamic light scattering (DLS), which measures the diameter, particle size distribution (PSD), and stability of NPs in suspension. This procedure has International Standards Guidelines at the nanoscale [48] but has contradictory results [63]. Its difficulties lie in monitoring minimum variations in diameter [100] and determining the size distribution, and it only works if the particles tend to monodisperse [101]. It is preferable to combine different techniques in a complementary way to evaluate the size, PSD, and stability [102].

Laboratories such as the NCL and EU-NCL are contributing to promoting the use of Standard Operating Procedures (SOPs) to determine the size distribution of NPs by coupling Asymmetric Flow Field-Flow Fractionation (AF4) with size measurement by DLS and/or Multiangle Light Scattering (MALS), obtaining the AF4-MALS-DLS combination. This technique applicable to nanomedicines could be an alternative for regulatory purposes [103], differentiating the particle population, discriminating aggregation, and monitoring batch-to-batch PSD changes or instability during long-term storage [104,105].

The strongest technique to know the morphology and size of NPs is electron microscopy (EM), which provides images with higher contrast, with elements of high atomic number, and a resolution of less than 1 nm [106]. There are different variants of this technique, such as transmission electron microscopy (TEM), which allows a large population of particles with high lateral resolution to be observed [107]; scanning electron microscopy (SEM), a technique that determines topochemical data and surface defects [108]; and the combination of both: transmission and scanning electron microscopy (STEM), through which the size distribution, interfacial structure, compositional distribution, and phases of the NPs can be identified [109]. However, with these methods, it is impossible to determine the presence of agglomerates, and it does not adequately identify the organic coverage of certain particles because a small, non-representative sample is evaluated [110]. TEM diagnosis cannot establish particle thickness [107], while SEM analysis causes electron beam degradation that alters or destroys particle surface details [108]. Atomic force microscopy (AFM) is another alternative to mass spectrometry (MS); it allows a three-dimensional analysis of the NPs, mapping the surface heterogeneity and identifying their thickness, without the need for sample preparation and electron beam incident [102]. The observation depends on the probe (tip), which limits the lateral resolution and leads to an overestimation of the dimensions. In addition, the sample must be prepared, which alters the particle conformation [107,108]. For regulatory purposes, and a proper evaluation of morphology and size, the regulations suggest combining several methods with different configurations [111].

Another of the most important parameters in the characterization of NPs is encapsulation efficiency (EE), a quality attribute which results in estimating the drug loading capacity in the particles [112]. It depends on the preparation process, physicochemical properties of the drug, and formulation variables [113]. It is possible to quantify this parameter through direct methods that evaluate the encapsulated drug and indirect methods that calculate the non-encapsulated drug [114]. The EE does not always reflect the exact percentage of the entrapped drug, and there are several factors to consider that can influence its determination, such as the synthesis conditions of the NPs or the concentration of the active ingredient [115]. The main challenge in calculating EE is the accuracy of the drug analysis, where there are complications in the use of both direct and indirect procedures, and a combination of both types is preferable to obtain more accurate results (drug-loaded solid lipid NPs were separated by centrifugation before measuring the unencapsulated drug in an aqueous phase, in other studies, the drug content was directly measured to determine EE%) [115].

Another essential quantification parameter in evaluating the properties of nanoformulations is their release kinetics. This characteristic should be studied by analyzing factors that can influence the process and identifying the drug’s release (in a slow and sustained manner or with a complete expulsion) [116]. This assessment depends on the nature of the encapsulating matrix, although the drug loaded affects the release pattern [117]. The release kinetics can be studied utilizing various mathematical models, but only if the system conforms to certain initial conditions specific to each equation [118]. The physiological environment of the administration site (pH or temperature) can accelerate or decelerate the release of the encapsulated molecule, and it is important to consider the environment to which the NPs will be exposed in an in vivo model (protein-based hydrogel biocomposites were pH-sensitive and their degree of swelling was significant at pH 7.4 and at a temperature of 37 °C) [119]. In vitro kinetics is a first approximation of drug release over time, yet sometimes it may not reflect the actual concentration of the drug released [120]. However, its determination predicts the escape phenomenon of the encapsulated active substance and provides information for in vitro–in vivo correlation studies (IVIVCs), recommended by regulatory agencies [117]. The best-fit mathematical model must be selected to properly analyze IVIVCs; the most complex ones use many parameters, and their fitting algorithms are not always accurate [121].

It is necessary to identify and evaluate the points that alter the formulation and depending on the NPs varies its presentation: suspension [122], gel [123], cream [124], tablets [125], aerosols [126], or nanocomposites [127]. For gel nanoformulations (chitosan gel containing acrylic-based nanocapsules), it has been demonstrated that the zeta potential of the NPs influence on the viscosity, rigidity, and gel network structure [128]. In the case of tablets (dexamethasone-loaded PLGA NPs embedded in alginate), the encapsulation efficiency, morphology, and diameter of the particles can affect the thickness of the formulation [125].

The complete characterization of NPs considers particle-related parameters (size distribution, morphology, zeta potential, composition, charge, and surface chemistry) and entrapment parameters (drug charge and distribution, encapsulation efficiency, release kinetics, and cell interaction). For the evaluation of these attributes, it is essential to use the most appropriate combination of techniques in each case (Figure 2).

### 3.2. Regulatory Aspects of the Physicochemical Characterization of Nps

Regulatory agencies frequently reject nanoformulations because they find fault with the physicochemical characterization, especially the use of non-validated and non-standardized methods, lack of documentation on the elimination of assay interferences, and lack of justification of critical product parameters [129]. The FDA guidance for products with NPs provides some quality attributes that should be described and measured, such as size, PSD, shape, and surface charge [101]. The FDA recommends using several methods [76]. In the case of copolymeric micelles and liposomal products, EMA and MHLW indicate parameters, but do not specify validated methods, so the assays to be used are at the investigators’ discretion [68,71,75]. The agencies have suggested that recommendations, protocols, and methods from standardization organizations (e.g., the International Organization for Standardization (ISO) and the American Society for Testing and Materials (ASTM International)) be taken into account [130]. Some examples are the specific technical report for the physicochemical characterization of products with nanoscale materials (ISO/TR 13014: 2012); size analysis by DLS (ISO 22412: 2017) [131]; size measurement using TEM (ASTM E2859-11(2017)) [76]; as well as the European Committee for Standardization (CEN) standards for medical devices with nanotechnology [25].

In order to support industry and academia in the development of these new technologies [64], the “Assay Cascade Protocols” defines a set of the most reliable techniques based on reviews with various characterization methods that researchers use in the design of nanoparticles products [132]. We present a summary in Table 2 with the mentioned factors and related methods researchers consider in the physicochemical characterization of nanoformulations.

## 4. Challenges for In Vitro Biological Evaluation

The safety profiles of nanomaterials and nanoformulations in drug delivery and therapeutic applications are of great concern to the scientific community. To evaluate the biological activity of both the encapsulated drug and the particle in an in vitro model [159], it is necessary to first assess the type of target cell [160], the route of administration of the system [161], the degree of cellular uptake, and the release kinetics of the active pharmaceutical ingredients (APIs) in the tissues [162]. Some fundamental evaluations, such as cell line and model selection, interaction with biological fluids, and toxicological evaluations, are detailed below.

### 4.1. In Vitro Assays: Cell Lines

In vitro models are used to study the biological responses triggered by therapeutic agents [163] and to identify the main pharmacokinetic obstacles of the candidates [164]. Cell line assays provide important benefits such as easy pharmacological manipulation and genetic modifications in lineages [163]. These systems test the biological activity and toxicity of nanoformulations, considering cellular deposition, cell barrier permeation, and cellular uptake, considering more complex exposure changes for NPs [165]. In vitro models lack the tissue or organ environment and cannot predict the biokinetic profile of a new formulation [166], leading to the misinterpretation of data [167]. Many cell lines do not predict tissue or organ-specific damage, rendering the system insufficient [168]. For example, in a study evaluating the toxicity of folic acid and PEG-functionalized silicon NPs using in vitro (rat glioma cell line (ATCC^®^ CCL-107™)) and in vivo (zebrafish larvae) studies, researchers found that the particles did not affect cell viability in the in vitro model. In contrast, research has shown indicators of toxicity (alterations in motility) in the in vivo system [169].

Other factors that alter and impede the experimental reproducibility of in vitro assays in nanomedicines are the colloidal stability of NPs in a cell culture medium [170], the NP-cell interaction [171], the method of administration of the nano-drug [172], and the interaction of NPs with the membranes of the culture system [173]. The pH, composition, and temperature conditions of the culture medium negatively affect the stability of NPs and lead to the settling of aggregates on the cell surface of the culture with a decreased transport rate [171,173].

Standardizing cell lines and culture media is important to minimize variations in in vitro experiments [174]. Models should mimic human tissue conditions, simulate dynamic changes (pH, salt concentration, temperature) [175], and generate accurate results [176]. The selected cell line should translate the functionality and physiology of the tissue, organ, or system for which the nanoformulation will act upon [168] and express the signaling pathway of interest, enzymatic profile, and quantitative determination of damage caused by environmental factors [177]. Standardizing these experimental variables should limit inter-laboratory variability and make the data generated more comparable between in vitro and in vivo assays.

When evaluating the effects produced by NPs in in vitro models, it is necessary to work with advanced cell culture systems: two-dimensional monolayer, co-culture, or 3D models [165]. These assays improve the physiological relevance of conventional culture due to their larger cell-to-cell contact area [178]. The model increases its predictive value when mimicking the physiological conditions of living tissue [177]. In an in vitro intestinal uptake and permeability study on hyaluronic acid nanogels in a human colon adenocarcinoma cells (Caco-2) monoculture and a Caco-2/HT29-MTX co-culture with mucus, results showed that adsorption and sustained release was higher in the co-culture with mucus production than in the monoculture [179]. The most commonly used in vitro models are two-dimensional monolayer cultures because of their cost, high yield, and reproducibility [168], but their main disadvantage is the lack of representation of the in vivo microenvironment [180]. As an alternative, Transwell systems, cell inserts in a permeable membrane, began to be used to study various metabolic activities in vitro [181]. These systems have been crucial for understanding the transport mechanisms of NPs, as they more realistically describe and mimic absorption, distribution, metabolism, and excretion (ADME) [175]. The method is more costly and complex due to variations in temperature, humidity; they require an optimized medium that is difficult to develop; they can take months to establish and are affected by phenomena that hinder the free passage of NPs [165]. The formation of particle aggregates can decrease transport across the membrane, retention in filters, or formation of cell multilayers or vice versa, increasing the transit if there are holes in the barrier or lack tight junctions between cells [173].

Design refinement in cell line assays is suggested for higher throughput, using new technologies such as microfluidics or 3D cultures [182,183], which provide a comprehensive understanding of transport and toxicity processes [184]. These models can predict the toxicity of the formulation, for example, in an experiment on the toxicity of silicon dioxide (SiO_2_) NPs in a three-dimensional system of lung cancer cells (A549, ATCC^®^ CCL-185) and mouse fibroblasts (L929, ATCC^®^ CCL-1), the analysis with 3D cultures found that the toxicity of the particles was dependent on the in vitro model used (two-dimensional or three-dimensional), so their selection was critical in the toxicological profile of the NPs [185]. Standardization of these experimental variables should limit inter-laboratory variability and make the data generated more trustworthy and reliable.

### 4.2. Characterization in Biological Fluids

Another indicator to be evaluated regarding in vitro characterizations is interactions with biological fluids. When the biopharmaceutical is administered in nanosuspension, it comes in direct contact with body fluids, and the nanoformulation may undergo modifications [186]. Dynamic environments due to variations in electrolyte concentration, pH, and plasma proteins induce changes in the morphology of NPs [48]. The dynamism of the physiological environment also influences the stability of NPs and determines how much of the drug, completely encapsulated, reaches the target site [187]. If the carrier has lost its integrity, the drug is released, and the dose at the site may not be sufficient to prove efficacy [188]. Therefore, biological characterization must be performed in a test system as close to the biological environment as possible [189]. For nanomedicines, it is advisable to use blood or other human fluids rather than enriched fluids that simulate biological ones [190,191]. There are some sensitive methods to use such as DLS, Field-Flow Fractionation (FFF)1 (ISO/TS 21362: 2018), Nanoparticle Tracking Analysis (NTA) (ISO 19430: 2016) [48], or more robust combinations such as AF4-MALS-DLS [192].

The main obstacle NPs have is lysosomal degradation when entering cells by phagocytosis, endocytosis, or pinocytosis. Particles must escape from these compartments to effectively concentrate on the active principle and perform their therapeutic activity in the correct cellular location [193]. It is possible to develop particulate systems that can escape from lysosomes by using different materials, such as chitosan, or by modifying their composition [194,195]. However, the development of safe and effective systems that can reliably deliver cargo to the cytosol remains a challenge [195], as some of the materials cause cell damage [196], and a balance between lysosomal escape and toxicity needs to be found [193].

Other reports describe the phenomenon of protein corona formation on NPs [48,197,198]. The event is based on the coating of resident proteins in a biological fluid that comes in contact with the particle surface, can interact with the surrounding medium, and depends on the physicochemical properties of the system and the nature of the biological medium [199]. The composition of this corona influences the biological activity of the encapsulated molecule and its release kinetics [200], as depending on the concentration and type of proteins present, the circulation time may increase or decrease (Figure 3). So far, researchers have developed specific SOPs to study the behavior of NPs in the presence of serum proteins to define their biological identity and understand the in vitro effects [48].

### 4.3. Toxicity in In Vitro Models

Understanding the mechanisms associated with nanotoxicity is essential for formulation application [201]. Some of the initial toxic effects induced by NPs include cytokine production, inflammatory stimuli, increased reactive oxygen, and nitrogen species, leading to apoptosis, necrosis and autophagy-mediated cell death mechanisms, and cytotoxicity [202]. The production of redox species (ROS) during mitochondrial dysfunction allowed for determining the pathological role of apoptosis in NPs toxicity [203]. Dysfunction of mitochondria leads to endothelial reticulum system (RES) stress, lysosomal dysfunction and impairs normal functioning with aggregation of unfolded proteins during cellular rescue mechanisms [203,204]. NPs can activate the inflammasome and extracellular trap formation in neutrophils, induce macrophage polarization and reprogramming by stimulating epigenetic changes, increase proinflammatory cytokine production and activate the complement system [205]. Cell damage caused by NPs is also due to non-oxidative mechanisms, such as reduced nucleotide, carbohydrate, amino acid, and energy metabolism [159]. Various methods can assess cell viability to predict toxicity before animal testing, thus minimizing its effect on a living organism [206]. There are four main methods: loss of membrane integrity, loss of metabolic activity, loss of monolayer adherence, and cell cycle analysis. The data generated by these various viability assays can help identify cell lines susceptible to nanoparticle toxicity and give cytostatic/cytotoxic clues to the location of cell injury. Some systems stimulate inflammatory responses, initiate oxidative stress, and cause DNA damage (genotoxicity) and cell death (cytotoxicity) [207], which greatly limits their use in diagnosis and treatment, and finding the right balance between therapeutic effect and adverse events is a problem [80]. Although the attributes that correlate with the toxicity of NPs are: the chemistry of the formulation material, the concentration, the size, and the type of cell-targeted [208]. Subsequent immune responses by internalization in macrophages occur in particles smaller than 100 nm, spherical with a large surface area, which adsorbs serum proteins [205,209]. Several strategies to reduce this uptake include synthesizing particles larger than 100 nm, with alternative shapes, and decorated with hydrophilic molecules on their surface [205].

There are many assays to evaluate the in vitro toxicity of NPs. We will consider only some of the most novel ones because each formulation has a different purpose and design; therefore, they must have situation-specific adjustments. For instance, CELLigence analyzes cytotoxicity and works as a non-invasive in vitro method that observes all cell growth events, tissue cells in real-time, proliferation kinetics, size, reproduction, and morphological effects [210]. It avoids the interaction of chemicals, dyes, and other cells, unlike other conventional cytotoxicity methods. Some nano-specific interactions can make the demonstration of toxicity even more challenging such as NP’s acting as inhibitors or enhancers in addition to absorbing or scattering light and even reacting with test reagents [211]. The best non-invasive imaging tool to identify numerous processes in a cell, such as migration, differentiation, and cell death, is digital holographic microscopy and its dark-field variant; it produces photographs with extended depth of focus and allows the rapid assessment of cell viability with dynamic or quantitative measurement of shape and volume with high sensitivity [212]. This tool is most useful for live-cell imaging, early cell death detection, cell permeability in fluids, and the toxin-mediated analysis of single-cell morphology [213].

The monitoring and regulation of ROS levels is an essential tool, and the prominent role of NPs in ROS production and its consequences recommends the evaluation of nanotoxicity (Figure 4) [201]. New and more accurate assays are now available such as Fluorescent Probes for ROS Measurement [214]; Genetic Approaches for ROS Detection [215]; Nanoprobes for ROS Detection [216]; and Nanoelectrodes for Measurement of ROS in Superparamagnetic Iron Oxide Nanoparticles [217].

NPs enter cells, react with cellular components, and remain in cells, resulting in long-term toxicity, with the essential determination of genotoxicity. There are no established guidelines for performing these tests. Most common tests for these measurements include DNA fragmentation and electron microscopy, the COMET Chip assay, flow cytometry/micronucleus, flow cytometry/γ-H2AX, fluorimetric detection of automated alkaline DNA unwinding (FADU), gene chips, and G-banding analysis [218]. Some methods required modifications when analyzing in vitro genotoxicity; for instance, a robotic system replaced the FADU assay providing more flexibility, easier handling, accurate reagent dispensing, complete light protection, and temperature regulation [218]. Another interesting assay is the ToxTracker reporter assay with high sensitivity, high throughput screening with a panel of six cell lines with an embryonic mouse stem (mES), and several green fluorescent protein (GFP) tags for unique cellular signals. It is considered a highly sensitive technique for detecting genotoxic and non-genotoxic substances that assesses the genotoxic potential of NPs [219].

In vitro models can be used for studying immunotoxicity pharmacokinetics (ADME) and immune responses, while enzyme-linked immunosorbent assays (ELISA), flow cytometry, and reverse transcription polymerase chain reaction (RT-PCR) are best suited to analyze cytokine expression. Due to these limited in vitro methods for predicting immunotoxicity, the full toxicology cannot be studied [220]. However, at this time, there are no particular regulatory methodologies to measure the in vitro immunotoxicity of NPs.

The same is true for carcinogenicity, many formulations are aimed at anticarcinogenic activity, but the probability of causing cancer is also high. Studies of nanotherapeutic products are inconclusive, and the database for assessing carcinogenic risk is deficient, considering its qualitative and quantitative effects.

Concerning the assessment of organic toxicity for nanoparticles, the liver is the organ of greatest concern and the most explored because conventional animal models are not adequate to accurately assess hepatotoxicity [221]. Numerous tests in this area stand out, such as the 3D Microfluidics technique to grow living cells or organs on a chip [222]; 3D Liver Bioprinting [223]; and lastly, 3D Organoid Scaffolds which allows cells to be grown three-dimensionally [221].

We can summarize that the specific evaluation of these systems to determine the in vitro toxicity of formulations presents several difficulties for in vitro assays as there are many differences in the in vitro and in vivo study designs (Figure 5). It is necessary to look at the overall in vitro model, from the cell lines, the exposure medium, the culture system, and the method of administration of the nanoformulation, so that the assay can be developed in a cell system very close to human in vivo conditions [175]. More evidence still needs to be gathered to support these studies.

### 4.4. Regulation of In Vitro Models in NPs

Researchers have built the evidence for NPs in cell line trials on the experience of the scientific community [165]. In vitro models for NP trials are an important ally, helping researchers to elucidate some aspects of the safety and efficacy of nanoformulations, particularly if no previous data are available [49]. According to the International Council for Harmonisation of Technical Requirements for Pharmaceuticals for Human Use (ICH) guidelines for biotechnology products, in vitro models (cell lines and primary cultures) are accepted to analyze the effect of the biopharmaceutical on cell phenotype and proliferation [224]. The regulatory environment for in vitro models in NPs is not clearly defined, and so far, no specific guidelines govern the selection of cell lines [47].

One of the guidance documents on in vitro assays and the selection of a suitable cell lines is the Organisation for Economic Co-operation and Development (OECD) guidelines for Good In Vitro Methods Practices (GIVIMP) [177]. This document reduces assay variability errors and demands confirmation that strains come from a stock authenticated by approved cell banks such as American Type Culture Collection (ATCC), Japanese Collection of Research Bioresources (JCRB), United Kingdom Stem Cell Bank (UKSCB), Deutsche Sammlung von Mikroorganismen und Zellkulturen (DSMZ), among others [225]. It includes required data and instructions on documenting the origin of the cultures, the strain, age, number of donors, isolation technique, etc. [225].

The Preclinical safety evaluation of biotechnology-derived pharmaceuticals (ICH S6) guidelines recommend ensuring a rapid regulatory process, and that product information includes criteria for selection of cell lines, culture media, assay reagents, and how interferences with each are eliminated [224]. This guideline states that it is acceptable to use mammalian cell lines and primary cultures to predict the effects of the new product on phenotype and its biological activity [224]. When choosing the cell model, the choice of strain should be justified, and the translation of the system’s functionality, physiology, and toxicity should be explained [75].

The EMA has approved 3D cultures that mimic the biological environment [226]. The standard for the assessment of in vitro cytotoxicity of medical devices (ISO 10993-5: 2017) has developed a screening method to assess the safety of NPs in 3D cultures (ISO/AWI TS 22455) [165], and provide guidance on the selection of cell lines in the biological evaluation of medical devices. ISO/TR 21624: 2020 guides nanomedicines using the inhalation route as culture, monoculture, and co-culture systems [178]. Specifications and documents that guide the step-by-step selection of cell lines and the most suitable system that translates the functionality, physiology and toxicity of NPs are still lacking. We hope that with technological advances and the support of academia, industry, and agencies, we will find the ideal strategy to overcome the gap in the predictive value of in vitro models.

Concerning immunotoxicity, there are normative for its evaluation. ISO/TR 16197: 2014 describes and collects useful in vitro and in vivo toxicological techniques, including ecotoxicological screening of nanomaterials. This standard can be used for early decision-making in research and product development, rapid information on potential toxicological/safety issues, and preliminary assessment of the produced nanomaterials. ISO/TR 21624: 2020 provides many exposure systems and in vitro cell-based methods used in studies that simulate the design of an inhalation toxicology investigation.

The ICH S8 guideline provides suggestions on non-clinical testing methodologies to identify substances that may be immunotoxic, which will assist in immunotoxicity testing decisions. This includes standard toxicity assays and supplemental immunotoxicity studies such as host resistance, macrophage/neutrophil function, natural killer (NK) cell activity, cell-mediated immunity, and T cell-dependent antibodies (TDAR) [227].

## 5. In Vivo Tests

Animal trials that simulate the condition targeted by the active substance serve to demonstrate basic science proofs-of-concept for developing and evaluating therapeutic targets [228]. Performing in vivo studies improves our understanding of the behavior of NPs in a living organism [229]. Exploring physiological functions and systemic interactions between organs requires a whole organism; no in vitro assay can demonstrate the interactions and dynamics of a living organism [230]. Nowadays, several living models are used to measure the impacts of NPs on organisms. Considering the animal species, a much more realistic and predictive measure of the effect of the compound is created and can capture the complexity of target engagement, metabolism, and pharmacokinetics required in the therapeutic drug [231]. They should include the effects of particles on various organs and systems, such as the liver, heart, kidney, and immune system [232].

The results obtained should be extrapolated to humans, but it is difficult to match the structural characteristics and functionality of NPs with the physiological differences between animal species [91]. For example, the blood plasma components of each species affect the body distribution and cellular uptake of NPs differently [135]. The relative amount of immunoglobulins (mouse, rabbit, and human) influence the levels of phagocytosis and clearance of NPs, resulting in their rapid clearance from the bloodstream [190]. The safety and therapeutic efficacy of nanoparticles can only be assessed by rigorous in vivo testing and based partly on the results obtained from physicochemical characterization studies and in vitro assays (Figure 6).

### 5.1. Selection of Animal Models and Test Parameters

NPs, unlike other biopharmaceuticals and small molecules, induce toxicity independent of their physicochemical characteristics. Their surface area size increases exponentially, and reactivity increases and causes severe side effects [233]. NPs characteristics, such as large surface area, high catalytic activity, and unique optical properties, give rise to several challenges that often result in inconsistent results [234]. Choosing the appropriate model is one of the most important issues in predicting human biological responses [207,235]. The biological activity of NPs is determined through in vivo assays taking into account suitable models [236], where different organisms are evaluated [237]. In vitro tests do not accurately predict in vivo effects and conceal undescribed risks [238]. Animal testing is the most favorable and reliable method, translating the formulation’s biological efficacy and toxicity before human use [239]. There is not much literature on the type of animal species selected for one nanoparticle that translates the intended effect for the evaluated biological activity. Correlation between in vitro and in vivo assays is another critical factor for compiling the cascade of studies that converge in evaluating the same parameters [240]. Correct in vitro–in vivo correlation allows the prediction of in vivo performance (Figure 7). Several pieces of research indicate that the most suitable approach to investigate correlation is through experiments that simultaneously analyze NPs in vitro and in vivo [241,242,243].

The main in vivo model used for estimating the biological activities of different nanomaterial systems and that have shown immense potential to describe the mechanism of biological actions of NPs are invertebrate models *Caenorhabditis elegans* and *D. melanogaster* and for vertebrate models *Zebrafish* and mammalian (mice and rat) models [244].

Another animal model (higher organism) that is used with advantages in in vivo assays with NPs is the pig (*Sus scrofa domesticus*) [245]. This mammal shortens the phylogenetic distance between the rodent and human models due to its similarity with the immune-logical and lymphatic systems [246]. Acute hypersensitivity reactions (HSRs) induced by intravenous (IV) drugs and other compounds represent an ancient, unresolved immune barrier. The swine model has been proposed by regulatory agencies for preclinical risk assessment of HSRs in the clinical stages of nano-drug development as predictors of adverse drug reactions (ADRs) and severe adverse events (SAEs). The porcine model of complement activation-related pseudoallergy (CARPA) is a classical one, which determines the immune reactivity of nanomedicines. It has also been used in safe infusion protocols for reactogenic NPs such as liposomal drugs (PEGylated liposomal prednisolone (PLP)), which can provoke HSRs, with an exacerbated and toxic response [247]. Another example is the administration of solid lipid NPs encapsulating nucleic acid, ONPATTRO^®^ (Patisiran), approved by the FDA and EMA [248]. However, as rodents differ in their metabolism and physiology from humans, swine, like no other animal model, faithfully reproduces the human organism [249]. Animal models should be selected considering aspects such as correspondence with the route of administration, dose, experimental design, physiological state, and the stability of the nanomaterial in biological media [250].

The small size of NPs gives rise to several questions about their distribution in different systems, and they cross the different barriers (pulmonary, intestinal, cutaneous, and placental) at the tissue level, causing possible accumulations at the systemic level [251]. Several in vivo studies analyze the distribution of NPs in different routes of administration depending on their properties, time, and concentration [252,253]. The route of administration should be chosen based on the accumulation of NPs in organs, target and nontarget tissues, barriers, and physiological changes in the body [254,255].

Dose quantification is demonstrated in the measurement of the physical properties that determine its carrying capacity [256], calculating the dose based on the mass of the particle, and its measurements (ng/mL and mg/mL) are presented along with the weight of the NP, drug, and complete nanoformulation, as well as the number of particles administered per dose and the surface area of the particles [257].

The absence of controls in the assays is a common challenge [258]. Alternatively, already known substances such as Triton X-100, cobra venom, and nanoformulations already approved by regulatory agencies serve as a comparison starting point since the effect of these substances is already known [259]. For example, Taxol^®^ and Doxil^®^ are used as positive controls in the induction of hypersensitivity reactions, complement activation, and anaphylaxis, while Abraxane^®^ is considered a negative control for immunotoxicity testing [260]. Nanomaterials that use labels to track their biological function also include controls to demonstrate whether or not the labels affect the formulation’s behavior [261]. Controls are also for APIs [76].

### 5.2. Types of Preclinical Studies for NPs

Developing a pharmaceutical product is a stepwise process involving assessing animal and human efficacy and safety information. The objectives of preclinical toxicity assessment generally include a characterization of toxic effects regarding target organs, dose dependence, relationship to exposure, and possible reversibility [262]. This information is useful for estimating a safe starting dose and dose range for human trials and identifying clinical monitoring parameters for potential adverse effects [263]. In the early stages of a formulation or drug candidate, research on the product’s safety is necessary to obtain the first information on its tolerability in different indication systems relevant for future decisions [264].

Preclinical toxicity testing should be performed in two animal species with prolonged treatment periods and multiple doses [265]. In the selection of the animal model, it is very important to minimize as much as possible gender [266,267] and age bias, as they have to be related to the time of disease onset in humans to reduce failures in the extrapolation of data (Figure 8). The models include rodent and non-rodent species with no physiological relationship [226]. The most commonly used rodent species for preclinical studies are mice and rats [268], and there are conflicting results when extrapolating to humans [254]. For testing the pyrogenic potential of NPs, these species are not good predictors because of their resistance to endotoxin [269]. The animal model selection is crucial since some species mimic better and more sensitively the response generated in humans [270]. The most suitable species to evaluate the biokinetics of NPs and the overall toxicity profile are non-human primates, which are closer to human physiology and genetics; however, they usually are not as accessible due to their high maintenance cost and ethically related issues [91].

The route of administration influences the choice of parameters and techniques for physicochemical characterization, in vitro characterization, and in vivo characterization of the nanoformulation [254]. Selection should rely on the accumulation of the NPs system in organs, target tissues, and non-specific tissues, barriers, and environmental changes of the biological system [255]. In the FDA guidance for nanomaterials in industry, a description of the route of administration aspects can be found [76]. For example, accumulation/translocation of nanomaterials should be considered for inhalation administration, and subcutaneous administration of NPs may increase sensitization to other allergens [76]. Specific dosing schemes for nanomedicines use large particle drug designs as their baseline while introducing modifications based on the NPs properties such as agglomeration states [255] and biodistribution data [257,271]. Transmission electron microscopy allows for calculations using the size distribution, surface area, and concentration of NPs [272].

The therapeutic index usually determines the drug dosage and, therefore, standardization through preclinical testing is complex. Specific guidance for dosing in nanomedicines is based on large particle drug designs, introducing modifications based on the properties, agglomeration states of NPs, and biodistribution data [255]. The drug delivery dose is the patient’s administered amount (mg/kg body weight or surface area). For NPs, it can be stated as the number of particles delivered; however, it is always necessary to consider the amount of drug encapsulated for proper comparisons in animals testing.

### 5.3. Pharmacokinetics (PK) of NPs

Pharmacokinetic studies of nanoformulations are also crucial to assess toxicity [273]. Pharmacokinetic studies concerning NPs are scarce, and the lack of guidelines for nanomedicines makes it difficult to evaluate this parameter [274]. PK comprises four processes: absorption, distribution, metabolism, and excretion [275]. In each, the area under the curve (AUC), clearance (CL), the volume of distribution (VL), mean elimination time (t_1/2_), maximum plasma concentration (C_max_), CL of the kidneys, and the mononuclear phagocyte system are quantified [276]. The rate and extent of absorption depend on the physiological environment and the properties of the NPs [277]. Nanoformulations cross physiological and physical barriers that selectively inhibit the flow of molecules affecting the bioavailability of NPs [278]. Size, surface charge, and shape greatly influence cellular uptake [241]. For example, smaller NPs have greater intercellular transport by follicle epithelia and if the surface charge is positive there will be greater transport in mucosal and epithelial cells [279]. Absorption is related to the route of administration and the properties of the NPs [274]. The absorption rate, penetration, size, and influence of ultraviolet (UV) light in the dermis should be evaluated [280]. In the oral route, the negative surface charge has a higher absorption at the gastrointestinal membrane [281], and in the small intestine, it is related to size [282]. The pulmonary route has a greater contact area favoring absorption [282].

Physicochemical properties influence distribution (interaction with biological barriers and proteins) [274], while composition (silica, polymers, proteins, metals, lipids), size, morphology, surface charge, and hydrophobicity impact biodistribution [283]. For example, silica NPs have a higher affinity for the lungs [284] and distribute better in the liver [285]. The size determines prolonged distribution; the smallest limit in NPs for renal filtration is between 5.5 nm and 10 nm [282]. Their properties at biological barriers such as immune system, epithelium and mucosa, and blood–brain barrier mark the distribution process [286]. Upon contact with biological fluids, the surface of the NPs is surrounded by proteins, forming a structure called protein corona that alters the size of the NPs by changing the surface charge and making it anionic [287]. To predict the NPs behavior and circulation time, the description of the concentration and type of proteins attached to the NPs surface is performed [47]. This evidence indicates the necessity of making protein binding corrections to predict the PK of nanobiopharmaceuticals [288] accurately. Protein concentration coupled with physiological change due to diseased tissue may prevent NPs from reaching the target tissue [91]. The complexity of diseases requires an understanding of the general and disease-specific barriers and properties of NPs [289].

The clearance of NPs is very similar to that of conventional drugs, and they have two main routes of elimination: renal and hepatobiliary filtration [290]. Sampling is performed in urine and feces, but also in cerebrospinal fluid, alveolar lavage, or tissue biopsy [263]. Several factors influence the clearance process, the size, the smaller ones are eliminated via urine, and the larger ones are eliminated by bile [291]. The Mononuclear Phagocytic System (MPS) in the liver and spleen is also involved in clearance [292]. Macrophages, at high doses, opsonize the NPs causing loss of most of the injected nanoformulation, making it the first route of elimination from the body [276,292]. It is necessary to remember that the proteins bound to the NPs increase their opsonization. Consequently, the NPs’ design should minimize the binding of proteins to their surface [293].

The PK of biopharmaceuticals designed with NPs is more complex; after administration and subsequent absorption (Figure 9), “nanometric and non-nano forms” persist in the biological medium [279].

To understand the pharmacology of NPs, it is essential to analyze and quantify: the area under the curve (AUC), clearance (CL), the volume of distribution (VL), mean elimination time (t1/2), maximum plasma concentration (Cmax), and CL of the kidneys and the mononuclear phagocyte system. A clearance occurs in each component of the kidney filtration system. Depending on the NPs’ type, these may remain in the bloodstream or undergo renal filtration from the blood in the glomerular capillaries. In addition, the renal structures, such as the glycocalyx, endothelial cells or glomerular basement membrane, can recognize and select NPs for filtration. Filtered NPs are transported to the proximal tubule, interact with proximal epithelial cells, and get reabsorbed. NPs not selected for renal filtration may interact with the renal tubular compartment after being transported from the efferent arteriole to the peritubular network.

### 5.4. Immunotoxicity of NPs

The interaction of the formulation with the immune system can result in different events: it does not recognize the particles as a potential hazard and excretes them by renal filtration; it recognizes them as a threat and eliminates them by phagocytosis; it recognizes them as a hazard and initiates an inflammatory reaction; or persistent inflammation is activated and fails to eliminate them, causing tissue damage [294]. It is equally important to perform these studies on in vivo models because the assessment of immunological effects in vitro is limited [165]. Understanding the immunological compatibility of nanoformulations and their effects on hematological parameters is now recognized as an important step in the preclinical development of nanomedicines since the occurrence of immunological adverse events is responsible for 15% of drug failure in early clinical stages [240]. Several challenges arise in the immunological characterization of nanoformulations, the main one being the contamination of the systems with endotoxins or lipopolysaccharide (LPS) [295]. The large contact surfaces of NPs facilitate the binding of this type of contaminant that can adhere to most surfaces [296]. Endotoxin contamination is responsible for many immune/inflammatory effects attributed to NPs [297]. The detection and quantification of LPS present difficulties because NPs interfere with traditional analytical tests [240]. These interferences consist of an inhibition of detection and increased assay sensitivity [54], caused by the particles’ optical properties or surface reactivity [298]. Another challenge related to endotoxin levels is depyrogenation [296]. Due to the complexity of NPs and their easy tendency to alter their properties, there is no effective technique for endotoxin removal [299]. There are several depyrogenation methods for nanoformulations: UV irradiation, ethylene oxide treatment, formaldehyde treatment, and autoclaving [300]. These methods result in aggregation and changes in particle morphology, stability, and size distribution, thus increasing their cytotoxic capacity [297]. In 2020, Zielińska et al. proposed filtration as an alternative method for most NPs, although it also presents problems, such as the clogging of filter pores due to formulation viscosity and particle size [299].

In some assays, the mechanism by which the particles trigger the observed immune response is not yet understood [53]. The complexity of NPs’ systems generates the necessity to use many tests to determine their hematological profile [259]. Experimental design, controls, NPs interference, and endpoints are recurrently omitted [241,301]. For example, sometimes, the induction of CARPA is simply not evaluated or overlooked during the preclinical stage of the nanomedicine development; however, this might result in hypersensitivity reactions in patients during clinical trials [241].

### 5.5. Regulation of In Vivo Assays on NPs

The toxicity of nanoformulations is one of the most important challenges limiting the clinical translation of NPs. Regulatory agencies ensure that any nanomedicine must demonstrate a rigorous safety profile based on multiple key factors, such as physicochemical properties and route of administration [302].

Developers should plan what data they need to collect, such as the most relevant parameters influencing nano-drugs’ short- and long-term toxicity [303]. The determination of the concentration and median lethal dose (LC50 and LD50), the lowest concentration that causes a noticeable effect on the organism (LOEC), and the maximum concentration at which no observable effect is present on an organism (NOEC) allow the safety of nanomaterials to be assessed [304]. Their assessment should also include a consistent set of data at the different organ and toxicological endpoints to evaluate the individual components and the complete formulation [305].

Even today, during formulated NPs risk assessment, they are seen as conventional complex chemicals [306]. However, the same agencies have pointed out that nanoformulations should not solely be analyzed from a conventional chemical point of view because they exhibit physicochemical properties that make their analysis more complex [302]. There have been several attempts to harmonize toxicological procedures using various initiatives (scientific opinions, guidelines, and regulations) such as ICH and OECD guidelines, ISO, ASTM and FDA, EMA, and MHLW guidelines [307].

The OECD states that each test should comply with Good Laboratory Practice (GLP) under the standard section on safety studies [225]. ICH M3 (R2), ICH S6 (R1), S8, ICH S4, and ICH S9 refer to safety tests that may apply to nanoparticles [224,227,308,309,310]. It is convenient to use the standardized assays for toxicokinetics of NMs (applicable to NPs) (ISO/TR 22019: 2019), the Toxicity Screening method for NPs in 3D cultures (ISO/AWI TS 22455), genotoxicity (ISO/TR 10993-22: 2017). The nanotoxicological classification system (NCS) and multi-criteria decision analysis, along with other relevant parameters, help determine the nanotoxicity of NMs [311,312].

#### 5.5.1. Regulation of Pharmacokinetic Studies

For pharmacokinetic regulation, ICH guidelines such as ICH S6 (preclinical safety evaluation of biotechnology-derived pharmaceuticals), ICH S3B (pharmacokinetics: guidance for repeated dose tissue distribution studies), and ICH M3 (R2) are used [224,308,313]. In addition, the FDA guidelines for the industry for liposomal products and the EMA and MHLW reflection papers on liposomes and micelles include some recommendations to consider in the PK of nanoformulations [178].

The NCL mentions that PK studies are performed at single and repeated doses, planning tissue distribution studies and comparing bioequivalence [314]. According to the NCL and agencies, the parameters to be monitored are the same for conventional drugs [314,315]. Other critical variables for small molecules, such as intrinsic clearance and volume of distribution associated with the unbound (free) drug fraction, should be included [276]. At the same time, depending on the purpose of the nanoformulation, agencies may require other data. For example, generic dermatological nanoformulations, creams, and emulsions agencies request evidence of blood/plasma pharmacokinetic parameters and related microstructure information [73].

Regulatory agencies require detailed studies to know the nano-drug’s precise disposition, analyzing the encapsulated, non-encapsulated and total drug (encapsulated plus non-encapsulated) [316]. In recent years, it has become necessary to include PK of the concentrations of bound drug (non-encapsulated bound to plasma proteins) and unbound drug (non-encapsulated that has not bound to plasma proteins) because non-linear protein binding changes the pharmacokinetic profile of the nanoformulation [317]. Overlooking protein binding and assuming it does not influence the profile constitutes a bias error. A clear example was the amphotericin B nanoliposome trials that used the ultrafiltration method and assumed that protein binding was independent of the formulation. In that case, they estimated the encapsulated only by subtracting the protein-bound amphotericin B from the total amphotericin B concentration [276]. Now, considering new alternatives is very important to avoid bias and the lack of a suitable identification method for nano-drug fractions [48]. The NCL has developed an isotope-based method called The stable isotope tracer ultrafiltration assay (SITUA) that helps identify each fraction and account for changes in protein binding behavior [316].

As an additional tool for conducting pharmacokinetic trials, regulatory agencies have recommended incorporating quantitative and rational approaches such as the pharmacokinetic/pharmacodynamic (PK/PD) model [318]. This qualitative tool characterizes the relationship between pharmacokinetics and pharmacodynamics in a time-dependent manner. Well-designed PK/PD modeling provides a better understanding of the exposure–effect relationship and allows them to obtain benchmark PKs to reach the maximum efficacy response with reduced toxicity [319]. This interactive process offers a rational approach in hypothetical modeling that can support the optimization of pharmacokinetic assays in nano-drugs [318].

#### 5.5.2. Regulation of Immunotoxicity

The immunotoxic assessment of NPs has no specific regulatory framework or regulatory guidance [25]. The whole test spectrum relies on the ICH S6 and S8 guidelines [265,320]. Researchers can consult ICH Section 8 when a nanoformulation contains a low molecular weight drug and ICH S6 when the formulation contains a biotechnology-derived product [321]. They can also consult ICH S4 for chronic toxicity testing, ICH S9 for anticancer drugs, and the multidisciplinary guideline M3 (R2) for non-clinical safety studies [308,309,310]. These formulations’ complexity and difference from conventional drugs raise questions about the approval strategy in immunotoxicity trials [322].

When standard toxicity studies (STS) do not provide sufficient evidence to demonstrate the formulation’s safety, it is necessary to perform immunotoxicity studies using criteria such as dose used and severity of effects [323]. They include studies of immune function in rodent or non-rodent species (T cell-dependent antibodies, 28-day study at daily dosing) [320]. However, there is a question of whether ICH S8 immunotoxicity assays are a reliable assessment tool for NPs [324]. For example, ICH S8 lacks guidelines for testing CARPA induction, hypersensitivity, inflammasome activation, and myelosuppression [241]. The EMA in reflection papers for intravenous liposomal products and micelles states that in vitro and in vivo reactogenicity studies will be performed [320].

The scientific community and agencies have recommended using ISO and ASTM standards [178]. The ASTM includes analysis of hemolytic properties, the standard test for colony formation of mouse granulocytes and macrophages, and a quantitative test method for the chemoattractant capacity of a nanoparticulate material in vitro (ASTM E56-2525-08-2013, ASTM WK60373). The ISO standards contain a standardized framework for the detection of immunotoxicity for NPs (ISO/TS: 10993-20). It is possible to assess in vitro immunotoxicity by detecting cytokine, interferon, phagocytosis, leukocyte proliferation, etc. In vivo assays contain adjuvant tests, antibody response assays (TDAR), and local lymph node assay (LLNA)/local lymph node proliferation (LLNP) [325].

The three important requirements for the agencies are endotoxin levels, sterility, and depyrogenation [265]. From a regulatory point of view, endotoxin content in drugs is measured in endotoxin units (EU) (100 pg = EU) [326]. Once the result of the formula is obtained, the nano-drug must comply: for systemic administration, a limit of 5 EU /kg/h, for cerebrospinal fluid 0.2 EU /kg/h [326]. The most widely accepted assay by agencies for detecting endotoxin in nanoparticles is the Limulus Amebocyte Lysate (LAL) assay [327]. This assay has several limitations and, to overcome them, ISO 29701: 2010 (under revision) clarifies what steps to take when NPs adsorb test components on their surface or interfere with the final assay reading [296]. Consideration should also be given to the correct type of LAL test or using two LAL assay formats, using at least two LAL assay formats and two NPs controls, and minimizing the risk of interference by using nanoparticle controls [241,296]. After data obtention, another robust assay such as the endotoxin detection assay based on ELISA technology (EndoLISA), might be necessary to confirm the results [327]. If there are discrepancies between the two tests, agencies accept other alternatives such as the monocyte activation test (MAT) or rabbit pyrogen test (RPT) [54,328]. The RPT only serves as a LAL validation tool, never as the sole assay, because it has limited sensitivity, long reading time, and high costs [296]. Another emerging method is the recombinant Factor C (rFC) activation method (highly sensitive and quantitative) [328], recently included in July 2020 in the tenth edition of the European Pharmacopoeia [329].

Agencies also regulate the sterility aspect of nanoformulations [330]. The antimicrobial safety of a nanoformulation requires a probability of no more than one viable microorganism in one million parts of the final product, i.e., a sterility assurance level (SAL) of 10^−6^ (dimensionless) [330]. The microorganisms can be quantified using the methods described by the NCL, such as quantitative microbial detection by sampling on millipore devices, agar plates on different media, and an assay for mycoplasma determination [325].

Integrated immunotoxicity testing approaches for chemicals and biotechnology products contribute to the design of these studies in nanoparticles. The regulatory aspects mentioned in this section help improve the ability to predict adverse effects. However, it will be necessary to increase our understanding and better identify the relevance of immunological tests and the necessary changes for improving their correlation to humans [331].

## 6. Process Manufacturing

Nanometric platforms require complex synthesis processes to produce NPs on a large scale [332]. Manufacturing nanomedicines is a challenging aspect since the unique properties of NPs hinder the production process [51]. High costs in unit operations, difficulties in scalability, difficulties in infrastructure, and quality controls make the production process a critical step [39]. The main challenge in the manufacture of nanoformulations is that their production by laboratory methods is not scalable to the industrial level [333]. The choice is to use unit operations capable of producing large volumes of the nano-pharmaceutical in less time and with a higher cost [334]. The methods used for the production of NPs can alter the properties and final stability of the product. These changes in architecture or properties lead to problems with the formulation’s pharmacokinetics, biodistribution, and efficacy [33].

To ensure batch-to-batch quality in the manufacturing process, optimization of the method, both of the process parameters and the encapsulated formulation setting, was needed [335]. Manufacturing processes should have batch-by-batch control throughout the process, with strict monitoring to allow continuous feedback when assessing product quality and reducing errors, ensuring reproducibility of physicochemical properties and biological activity (Table 3) [306]. Klein et al. (2019) performed quality controls on the manufacture of chitosan NPs associated with human immunodeficiency virus (HIV) antigenic peptides by the ionic complexation method (nine batches with active principle and four control batches) [99]. The analyses showed a high degree of uniformity in size and variability of 16% per batch, showing that the production method used was applicable and reproducible on a large scale [99].

Production scale-up influences physicochemical characterization, biological activity, biodistribution, and pharmacokinetics [33]. When scientists compared the clinical activity of Doxil^®^ (PEGylated liposomal doxorubicin) with its bioequivalent (Lipodox^®^), they found the latter one to have a reduced efficacy compared to Doxil^®^ [344]. These differences could be due to the synthesis and scaling methods used in each case, with the result that two formulations with the same composition, manufactured in different pharmaceutical industries, did not exhibit the same efficacy [27,33]. At present, there is no defined combination of techniques used to monitor all the properties of batch NPs, nor is there a reference material to establish the comparison [345]. With increasing technology and experience gained, the industry has production methods aimed at reducing the unit operations in the synthesis of NPs (Figure 10), being more reproducible and controllable, and reducing the obstacles in the manufacture of nanomedicines [346,347,348,349].

### 6.1. Stability

Stability is one of the critical aspects to guarantee the safety and efficacy of the formulation since alterations of the product during manufacturing, transport, or storage can increase its toxicity [334]. This parameter is dependent on the dosage form from the physical point of view and drug-specific from the chemical perspective [350]. PSD’s homogeneity and drug encapsulation in equal amounts serve to evaluate the first approach [334]. The major difficulty that causes the physical stability of the formulation is agglomeration [351]. It is advisable to perform agglomeration control during the production process to maintain the stability of the nanoformulation [334]. The chemical surface of the NP influences the stability with respect to the size, morphology, and composition of the NPs, so the chemical stability should be evaluated, checking the integrity of the encapsulating matrix and the non-degradation of the active principle [352,353]. Another important aspect is to always think about during the nanoparticle design is its stability during the manufacturing process; otherwise, some problems will arise [286]. Controlling stability during the production process increases the success of the NP during in vivo assays [354]. Researchers have also found that formulation change or the addition of chemical stabilizers that modify surface properties avoid in vivo aggregation [355].

### 6.2. Scalability

Scaling up the production process for nanomedicines is another challenging aspect because the initial desired properties of the NPs (size, shape, zeta potential, or surface characteristics) can be lost or modified, leading to safety and efficacy issues of the nanoformulation [356]. These changes might occur due to complexity in manufacturing or failure at industrial versus laboratory scale [357]. In a nanocapsule (NC) study using the emulsification-diffusion technique, necessary scaling parameters were defined, and it showed that increasing the impeller speed and stirring time changed the size, with no variation in entrapment efficiency [333]. With increasing technology, successful scale-up was possible for some commercialized nanomedicines [356,358,359], and there are new successful methods such as high-gravity technology [360], high-pressure homogenization (HPH) [361], and microfluidics-based processes [335].

Quality controls should check whether scaling directly or indirectly affects the characteristics of the NPs [39]. Good sampling management is required (batch per batch and at the proper times to avoid sedimentation of larger particles or dissolution of smaller ones), and using the same characterization techniques, individually or in combination [362]. Defining the CQAs parameters during scaling provides a confidence interval in the industrial characterization [306]. In a report on the effects of critical process parameters (CPP) on the variation of CQAs (particle size, PSD, and zeta potential) of Flurbiprofen (FB) Nanosuspensions synthesized by High-Pressure Homogenization (HPH) technique, they concluded that adjusting the CPPs produced an effective and stable nanosuspension [363]. The definition of the CQAs limits strengthens the scaling process, as long as it is taken into account that the ranges established in the laboratory have to be re-optimized during each process due to the differences in the manufacturing techniques used at the industrial level [335].

### 6.3. Regulations in the Manufacturing Process

The production process of biopharmaceuticals should rely on Good Manufacturing Practice (GMP) standards [364], ICH, and the systematic approach Quality by Design (QbD), consisting of two branches: Quality Assurance (QA) and Quality Control (QC) [365,366]. The guidelines used for the production process are the following: ICH Q7 (API GMP) [367], ICH Q8 (pharmaceutical development and quality principles in design) [271], ICH Q9 (quality risk management) [368], Q10, and Q11 (quality controls and development of pharmaceutical substances) [369,370]. QbD is based on determining the final quality characteristics that the nanoparticles should contain, also called Quality Target Product Profile (QTPP), from the formulation approach to the manufacturing scale-up process [371]. This systematic QbD approach emphasizes understanding and controlling products and processes based on sound science and quality risk management [366]. ICH Q9 indicates that QbD starts with the definition of risk factors: CPP, quality controls, raw materials with their critical material attributes (CMA), and CQAs [368]. Once described, the CQAs, CPP, and CMA, a design space focused on the relationship between the characteristics of the nanoformulation and the production process, is ready [366]. Next is the establishment of the control strategy and plans for continuous monitoring. Its applicability has evidenced the successful fabrication of lipid-based nanosystems, anticancer polymeric NPs [372], and monoclonal antibody carriers [373]. These examples have demonstrated that QbD contributes to better design, reduced manufacturing problems, and less regulatory control [374]. Monitoring each approach’s component is required as the CQAs of nanoformulations are more sensitive and vary with critical attributes of the nanomaterial, raw material, and critical process parameters [375]. In 2016, Pfizer industrialized ACCURINS^®^, an antitumor polymeric nanoparticle, and made a description of the use of the QbD principle as a support in the design of this product, an effective principle to take it to its clinical use and commercialization [376]. It is recommended to confirm that the raw material complies with the label “generally recognized as safe substances” (GRAS) [374]. Under these parameters, regulatory agencies consider that a complete manufacturing process should include the following features: industrially scalable; reproducible per batch; quality controls at various points in its development; process improvements with technology; low production cost and high quality; and robust quality control systems [377].

Although the QbD approach promises an increase in the translation of nanomedicines, there are several problems in industrial practice [378]. The most common concern in nanomedicines for applying QbD is determining which CQAs to use [379]. In late 2018, the Association of American Physicians and Surgeons (AAPS) Guidance Forum for Industry on Biologics with Nanomaterials, FDA, noted that, as a reference, the process of defining CQAs for biologics and describing risks is an iterative process building on the knowledge of the molecule [373]. In the last decade, CQAs have been determined by risk estimation based on preclinical model assays and multicomponent statistical techniques for lipid and polymeric NPs [380,381,382].

All the requirements mentioned above apply to any drug, but the difference is that certain specifications are more complex [377]. In January 2021, the FDA committed to collaborating with the National Institute of Standards and Technology, U.S. (NIST) to improve the manufacturing processes of new technologies [383]. With these partnerships, the agencies and the scientific community will work collaboratively to increase knowledge about method optimization, scale-up, and quality control to produce solid research on the regulatory program in the manufacture of nanomedicines [383].

## 7. Clinical Trials

At the clinical stage, nanoformulations are the main cause of the failure of treatments that showed promise in preclinical trials [365]. Key issues related to clinical development include biological challenges, large-scale manufacturing, biocompatibility and safety, intellectual property, government regulations, and overall cost-effectiveness compared to current therapies [39]. Extrapolation of data obtained in animal models to humans has always presented a challenge for translation to clinical trials and is even more challenging for biopharmaceuticals with NPs [51,56]. There is a regulatory need for validated, sensitive, and standardizable assays that incorporate in vitro, ex vivo, and in vivo protocols to adequately assess the nanotoxicology of NMs during the early stages of clinical development [384]. Physiological, biochemical, and proteomic differences, disease heterogeneity, PK, and bioavailability are different in humans and animals, which makes estimating nanoformulation data more complex [385]. In a clinical trial of a vaccine candidate mRNA-1273 encoding the Severe Acute Respiratory Syndrome Coronavirus 2 (SARS-CoV-2) spike protein, the authors indicated that it is challenging to have no correlates of immune protection between animal and human trials [386]. Differences in the anatomy and/or physiology of animal species compared to humans must be established based on the different routes of administration. Preclinical studies of NPs should be conducted with appropriate randomization and blinding to reduce bias and evaluated with appropriate controls, including standard gold treatment and not just free drug solution [162]. Regulatory authorities suggest that the sponsor evaluates any changes in the manufacturing process, or at any stage of its clinical development, related to the drug product or its formulation [387] to determine whether the changes impact the product’s safety (Figure 11). Another difficulty at this stage is the clinical study design where the choice of the appropriate study size, the number of controls, and the timing of therapy administration are challenges whose correct choice determines whether or not the trial is relevant to test the study hypothesis [388]. Patient selection criteria, dosing regimen, the timing of therapy administration, stages of the condition, and duration time add to the challenges likely to contribute to failures in the translation of NPs therapies [39,365].

### 7.1. Clinical Trial Regulations

Regulation of NPs is under the control of each country’s regulatory authority (i.e., FDA, Therapeutic Goods Administration (TGA), and EMA). The regulations for nanoparticles in clinical trials are not specific for this type of drug and must follow the same rules as conventional drugs, and there are only certain specifications given that come from previous experience from the researchers [7]. There is a need for validated regulatory standards and protocols, specifically for nanoparticles that unite drug and medical device regulations. These protocols should consider the complexity, route of administration, pharmacokinetics, pharmacodynamics, and safety profile of NPs and provide information on clinical trial design and patient selection [303]. Excessive regulation is not required, which can affect the advancement of products in the marketplace, increasing costs to achieve regulatory approval and/or consuming a significant portion of the life of a patent. The lack of clear regulatory and safety guidelines has affected the development of these products towards timely and effective clinical translation [59]. For example, the scientific community has extensively investigated polymers as an effective platform for NPs strategies. However, their safety and efficacy are highly dependent on polymer molecular weight, polydispersity, molecular structure, and conjugation chemistry, and there is an urgent need for an appropriate regulatory framework to aid in evaluation [303].

Nanomedicine commercialization depends on several regulatory factors based on government manufacturing practices, quality control, safety, and patent protection [389]. The development of global regulatory standards for NMs must be established together in key countries interested in investing in these drugs. Although there have been great advances in the field during the last five years, closer collaboration between regulatory agencies, academia, research, and industry is still needed [25]. This area still needs new analytical tools and standardized methods to evaluate the physical characteristics of NPs: morphology, size and its distribution, area, chemistry, charge density and surface coating, porosity, hydrophobicity, with the impact of these parameters on the results of in vivo studies [201]. Regulatory authorities should work together to develop appropriate standardized test methods and protocols for toxicity studies and regulatory requirements, which will be necessary to ensure the efficacy and safety of current and future NPs [390].

### 7.2. Regulation for NPs in Latin America

Regulation in Latin America for nanomedicines has the same deficiencies as in the rest of the world, and even its information is more limited [391]. This problem arises because international regulatory agencies such as the FDA and EMA rule over national agencies, and the fact that development conditions are different delays this type of product [391]. Latin America has yet to establish a regulatory framework for nanomedicines, mainly due to the government’s lack of interest in regulating new technologies and the economic state of Latin America [392]. The Pan American Health Organization (PAHO), together with national authorities, has proposed the Pan American Network for Drug Regulatory Harmonization (PANDRH) initiative that supports pharmaceutical regulation in the Americas within the framework of national and sub-regional health policies, recognize pre-existing asymmetries [393]. PANDRH contributes to harmonizing regulatory aspects of safety, quality, and efficacy for medicines of chemical origin but does not mention specific aspects for nanomedicines or NPs, like the OECD and the International Pharmaceutical Regulators Programme (IPRP) [394]. PAHO/PANDRH and some entities such as the Agência Nacional de Vigilância Sanitária (ANVISA-Brazil), the Federal Commission for the Protection against Sanitary Risk (COFEPRIS-Mexico), Centro para el Control Estatal de Medicamentos, Equipos y Dispositivos Médicos (CECMED-Cuba) and the Administración Nacional de Medicamentos, Alimentos y Tecnología Médica (Argentina) belong to the IPRP (https://www.iprp.global/members (accessed on 11 December 2021)). There are a series of general initiatives proposed by Brazil, Mexico, and Argentina, which are the Latin American countries with the greatest regulatory strength. Brazil formed the Working Group on Manufactured Nanomaterials (WPMN), identifying the need for new guidelines and points of improvement and the inadequacy of existing guidelines for evaluating nanomaterials [392]. Argentina encourages nanomedicine through the provisions specified in the official gazette (Disposición 9943/2019). Mexico, as part of the Nanoscience and Nanotechnology Network (RNyN), created the initiative of a National Nanotoxicological Evaluation System (SINANOTOX) [395]. However, there is no specific basis for developing and implementing a regulatory framework for nanomedicine. Risk assessment approaches are still incomplete and international regulations that are feasible to apply to the region continue to be used [396].

## 8. Conclusions

Nanoscience has been progressing for decades, but its greatest splendor was evident in the late 20th century with the beginnings of biomedical applications for nanotechnology. It is an interdisciplinary area of science located at the level of millionths of a meter (10^−9^ m) in size and involves structures, devices, and systems with novel properties and functions due to the arrangement of their atoms with impact on our lives. Their development and application have created new challenges for the scientific community, industry, and regulatory agencies. The wide range of applications of nanomedicines requires proper characterization, and their properties must be interpreted to decrease toxicity. Despite the extensive development of nanomedicines, only 50 formulations have reached the market, so only a few encapsulant systems have been approved by regulatory agencies for therapeutic use.

The nanoscale size and large surface area make these systems suitable platforms for accessing previously unattainable therapeutic targets, with action at the tissue or cellular level in an exclusive manner, but they may present undesirable effects that limit their safety. There are numerous challenges in developing NPs and choosing NMs, as encapsulant systems aimed at achieving prolonged-release or enhanced pharmacological efficacy and/or therapeutic effect compared to standard drug treatments. These difficulties at all developmental stages are due to the lack of a clear and absolute international regulatory definition of these materials, their toxicity levels, and how they should be approached and explored. For example, researchers face challenges in formulation, physicochemical characterizations, or when used as a drug product, in cargo analysis, pharmacological stability, release, drug conjugation, selection of appropriate in vitro and in vivo models that correlate results, and quantification and evaluate biological activity. Other challenges arise due to regulatory gaps that lack essential data on the manufacturing process, pharmacokinetics, pharmacodynamics, pharmacodynamics, and immunotoxicity of nanomedicines, which delays their clinical translation, difficulties reported on numerous occasions. For this reason, many nanoformulations fail to achieve success in preclinical trials, and as a consequence, there are few trials in clinical research facing numerous regulatory challenges.

Although there is great expectation for nanomedicine and its importance in the pharmaceutical industry, there is little regulatory guidance in this area. At every congress, symposium, or meeting of the research community aimed at nanoformulations, there are several requests for consensus on developing these products. There has been a general lack of specific protocols to characterize these nanomedicines at physicochemical, biological, and physiological levels, which in some cases could have been responsible for a failure in the late stages of these products’ development. Even though there are numerous documents about it, there is still a lack of specific standardized guidelines that provide legal certainty to manufacturers, researchers, quality controllers, auditors, regulators from both the medical and public point of view and guarantee safety and satisfaction in their application. This uncertainty created by the lack of consistency across the board can ultimately hurt funding, research, and development, thus destroying public acceptance and perception of nano products.

For that reason, with this manuscript, we wish to unify the different challenges described in the formulation of human-targeted nanoparticles and regulatory regulations. The document describes each challenge separating each of the stages of development of nanomedicines and complements each section with the regulations, protocols, guidelines, guidelines, regulatory standards, etc., described to date for each section. We have not been able to locate in the literature a similar document that can completely orientate researchers who develop this type of biopharmaceutical. Although some authors have addressed the subject, it is not yet unified in a single text, with the difficulties or the regulatory scope appearing in separate documents.

In developing these structures, researchers must minimize complexity and consider dosing for human use for the formulation to have adequate efficacy as a clinical treatment. Reduce complexity to the minimum required to generate clinically translatable nanometer-sized therapeutics.

The lack of formal regulation of nanomedicines and nanomaterials for health-related applications is a global problem that somewhat delays the development of nanomedicines. Despite numerous attempts to consolidate regulatory information to provide a better development perspective for NPs in the biomedical area, there are no specific guidelines or regulations for these drugs. In addition, there are disagreements among evaluators about the evidence presented when approving an encapsulated formulation. This problem is present in large regulatory agencies causing important repercussions in approving formulations in developing countries such as the Americas, which guide themself by the international regulatory framework. Small molecules are often not licensed globally; for this reason, the nanomedicine community requires urgent consistency across the governmental sector to allow development to continue in line with expectations. Nanomaterials are not new, and the need and urgency for treatments for some diseases or conditions cannot be met with the current regulatory structure. Criteria must be unified, and formal guidance issued to the research communities. Institutions and governments have invested millions of dollars in the progress of nanomedicine over the last two decades, but clear guidance from regulatory agencies is urgently needed. We know that evidence has been building up from its development, but strength is required for products to reach the market with proven safety and efficacy.

## Figures and Tables

**Figure 1 pharmaceutics-14-00247-f001:**
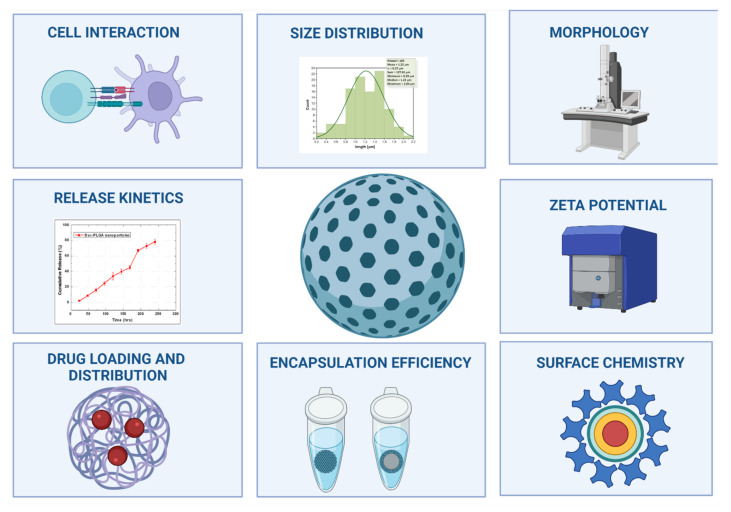
Main parameters to evaluate in the physicochemical characterization of biological NPs. NPs are complex structures that must be well characterized to identify how their qualities influence the quality, safety, and efficacy. Created with BioRender.com (accessed on 10 December 2021).

**Figure 2 pharmaceutics-14-00247-f002:**
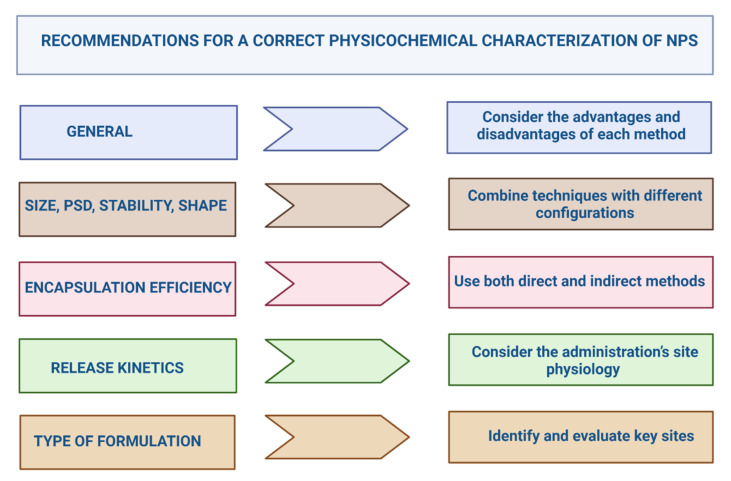
Recommendations for the correct physicochemical characterization of nanoformulations. Particle Size Distribution (PSD). Created with BioRender.com (accessed on 10 December 2021).

**Figure 3 pharmaceutics-14-00247-f003:**
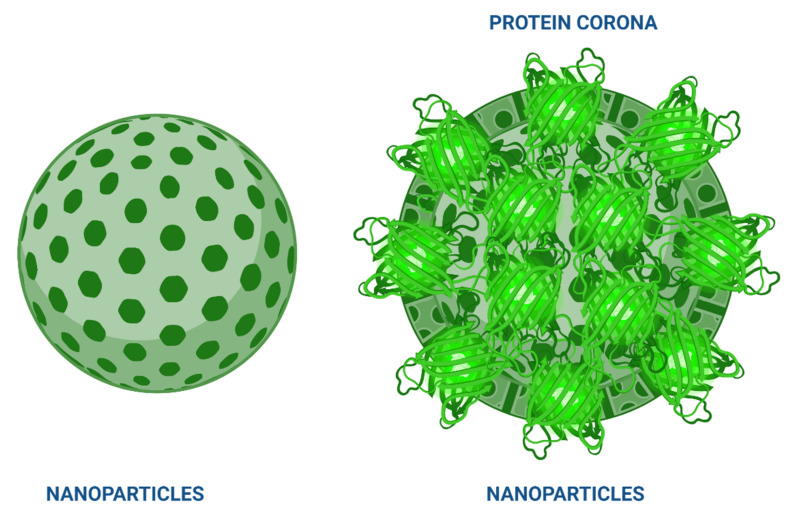
Protein corona formation on the surface of NPs. The corona effect occurs when nanoparticles enter the physiological environment. Protein corona and nanoparticles interact, and the adsorption of some proteins produces innate immune reactions such as complement activation and phagocytosis. Later, immunotoxicity reactions occur in some organs and systems. It is also possible to improve NPs’ behavior in the organism by regulating the corona proteins by appropriate surface modification and NP selection. Structural studies with NPs have demonstrated how to manage this biological phenomenon. It is even possible to design and construct special nanomaterials or functional groups and molecules that allow passage through the protein corona. Thus, protein-nanoparticle corona complexes provide design insights for new forms of encapsulation with NPs. Created with BioRender.com (accessed on 10 December 2021).

**Figure 4 pharmaceutics-14-00247-f004:**
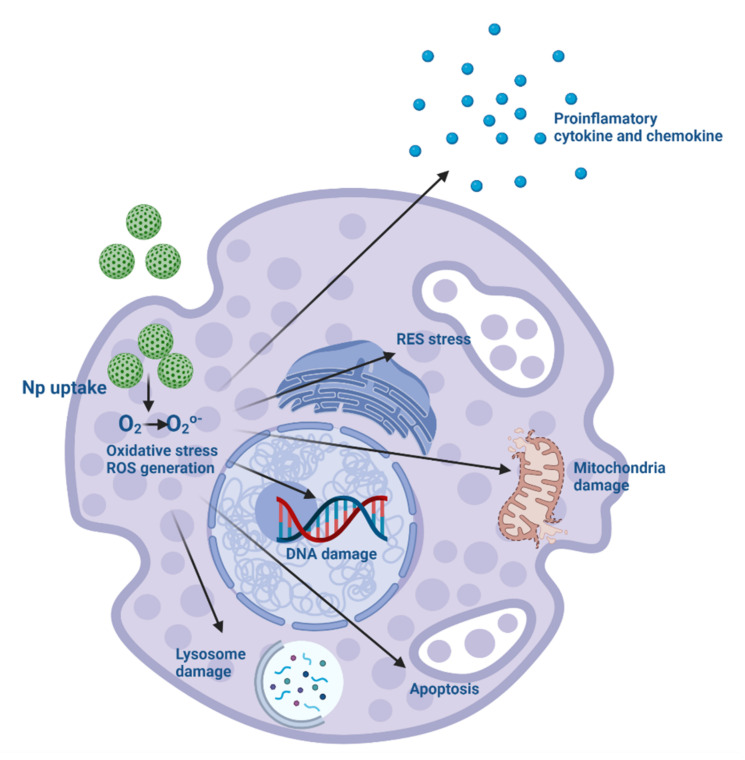
ROS generation caused by NPs, O2 O2o-. Reactive oxygen species (ROS) are natural subproducts of cellular oxidative metabolism that play an important role in modulating cell survival, cell death, differentiation, cell signaling, and production of inflammation-related factors. Altered redox homeostasis leads to damaging oxidative effects on cells mediated by interference with signaling mechanisms or biomolecules (proteins, lipids, and nucleic acids). NP-induced ROS generation initiates a sequence of pathological events, including inflammation, fibrosis, genotoxicity, and carcinogenesis, modulated by the physicochemical characteristics of the particles, such as size, charge, surface area, and chemical structure. This can generate toxicity with the expression of proinflammatory cytokines, fibrotic, activation of inflammatory cells (macrophages and neutrophils), which in each NP should be characterized and investigated. The mechanisms vary between particles, and the central cellular mechanism related to ROS production remains unexplained. Most NPs can cause toxicity facilitated by the release of free radicals. Created with BioRender.com (accessed on 10 December 2021).

**Figure 5 pharmaceutics-14-00247-f005:**
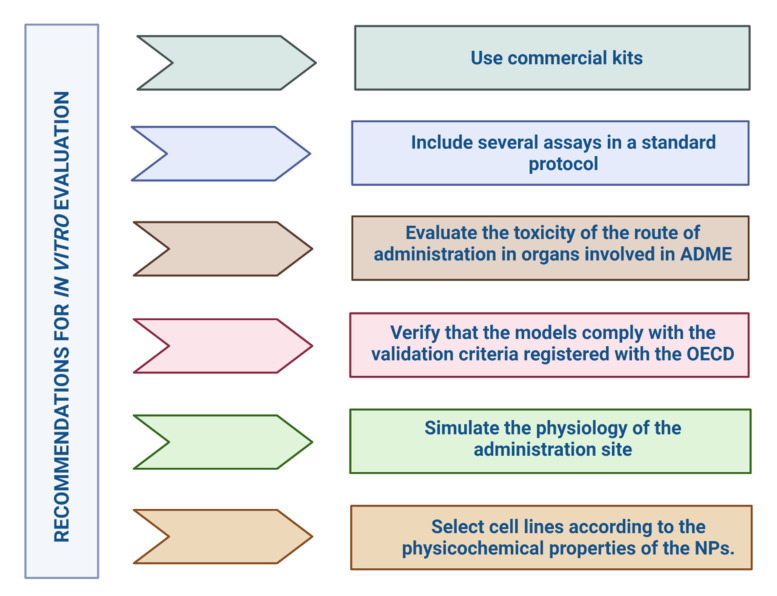
Recommendations for evaluating NPs in in vitro systems. ADME: Absorption, Distribution, Metabolism, and Excretion; OECD: Organisation for Economic Co-operation and Development. Created with BioRender.com (accessed on 10 December 2021).

**Figure 6 pharmaceutics-14-00247-f006:**
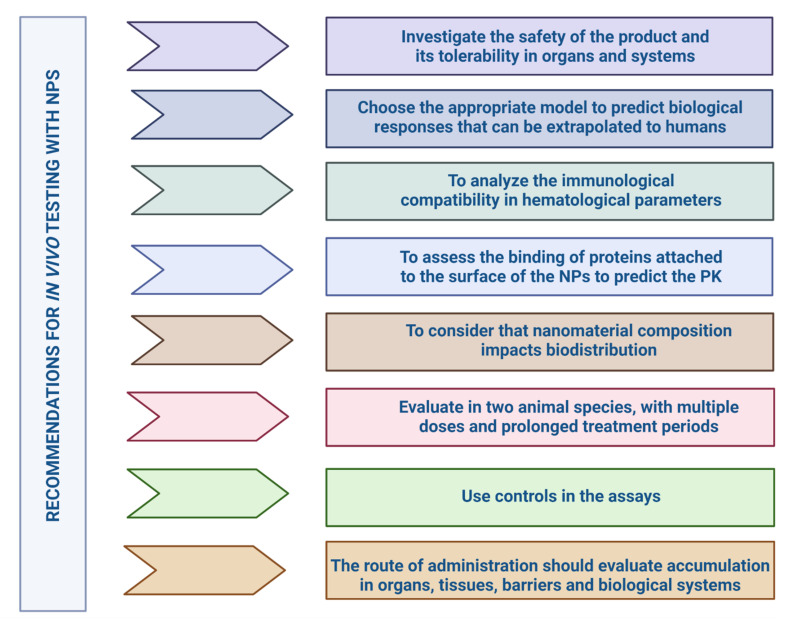
Recommendations for evaluating NPs in in vivo systems. PK: Pharmacokinetics. Created with BioRender.com (accessed on 10 December 2021).

**Figure 7 pharmaceutics-14-00247-f007:**
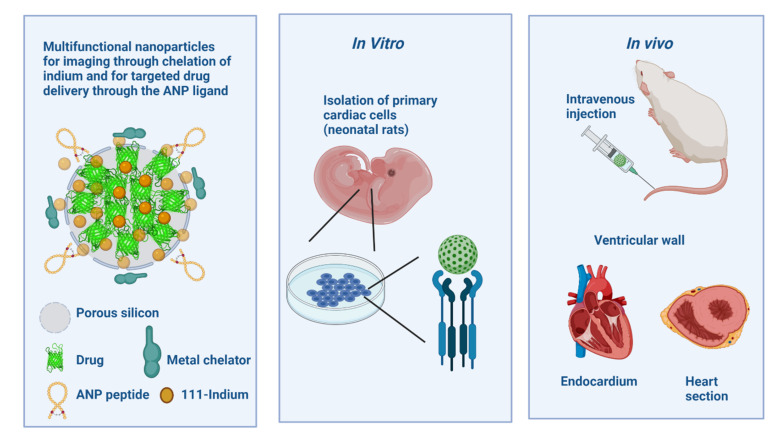
In vitro–in vivo correlation as a predictor of in vivo performance. Preclinical characterization of new nanotechnology-based formulations presents several challenges at different stages of their development: physicochemical characteristics, sterility, safety, and efficacy. The generally low sensitivity of standard in vivo toxicity tests to immunotoxicity, the interspecies variability in immune system structure and function, the high costs and relatively low throughput of in vivo tests, and ethical concerns about the use of animals, underscore the need for reliability in vitro assays. Correlation of in vitro and in vivo studies is required to ensure a more reliable transition to preclinical and clinical use. The figure shows cytocompatibility and cell-nanoparticle interaction. This method is very reliable for investigating the biocompatibility, biodistribution, and intramyocardial localization of cardiac-targeted NPs. After obtaining the results, the NPs can deliver a new molecule drug and evaluate its function in an animal model, thus establishing in vitro and in vivo correlation as strong evidence for further formulation development. ANP: Atrial natriuretic peptide. Created with BioRender.com (accessed on 10 December 2021).

**Figure 8 pharmaceutics-14-00247-f008:**
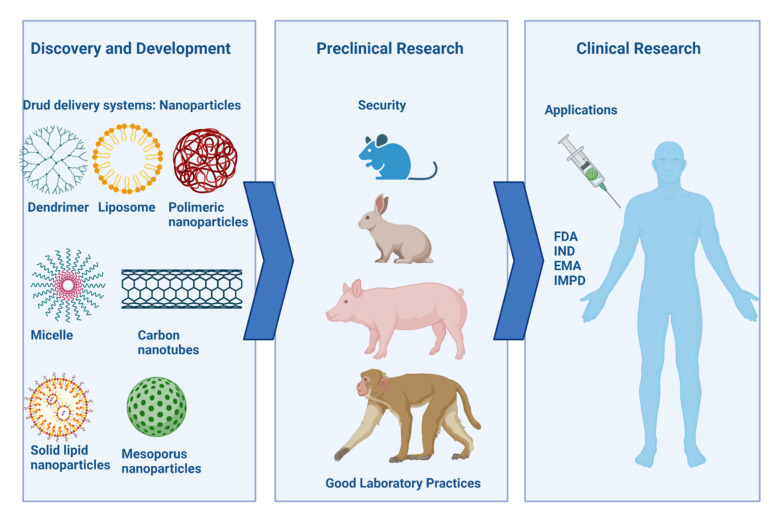
Characterization of preclinical studies for NPs in biomedicine. The common goal of preclinical studies is to identify potential immunological responses or other toxicities that can cause organ or system damage prior to administration in humans. The likelihood of characterizing NP toxicity increases as the formulation process advances from in vitro studies to preclinical in vivo models and clinical phases. In vivo toxicology studies include analysis of lymphoid organ weights, histological evaluation of immune organs and tissues, and understanding clinical chemistry parameters and hematology in two animal species: rodents and non-rodents (higher organisms). In addition, all these studies must comply with Good Laboratory Practices to be considered strong evidence by regulatory agencies. Extrapolating the results from these in vivo toxicity tests to humans is often challenging because of the differences in composition, organization, and sensitivity to certain agents between the human immune system and the one from animal species. The high cost of in vivo testing, and ethical concerns about animal use, limits the application of these experiments, despite their advantages in terms of toxicological predictability. FDA: US Food and Drug Administration; IND: The Investigational New Drug; EMA: European Medicines Agency; IMPD: Investigational Medicinal Product Dossier. Created with BioRender.com (accessed on 10 December 2021).

**Figure 9 pharmaceutics-14-00247-f009:**
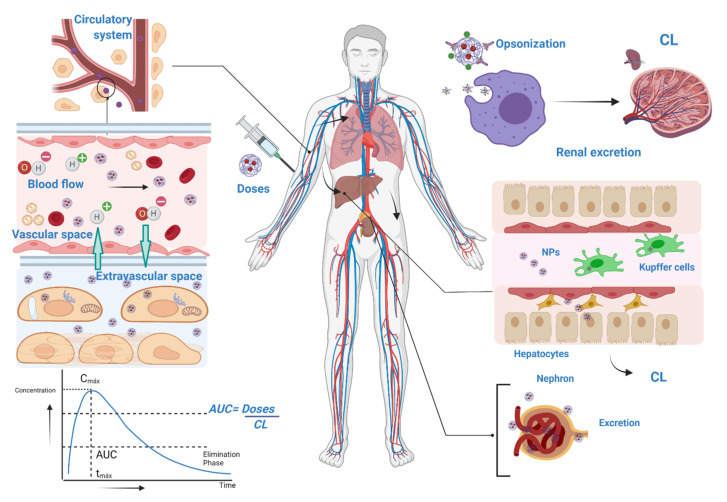
PK parameters for nanomedicines. Analyzing pharmacokinetics as a whole is a key component to understanding the biodistribution of NPs. Nanoparticles enter cells by passive transport such as red blood cells and directly depend on the size of the NPs, being only possible for sizes below 200 nm. Opsonization begins once the NPs enter the organism, and opsonins bind to the NPs and present them to the mononuclear phagocytic system (MPS). The rapid response of the MPS and the reticuloendothelial system (RES) results in the elimination of NPs within a few hours after injection, and they do not reach the target tissue. The uptake is dose-dependent, and it is challenging to understand the effect of dose on the system. Performing a quantitative biodistribution allows the assessment of their distribution in organs and tissues of interest, regardless of the route of administration. The largest accumulations of NPs occur in the blood, liver, and spleen. Created with BioRender.com (accessed on 10 December 2021).

**Figure 10 pharmaceutics-14-00247-f010:**
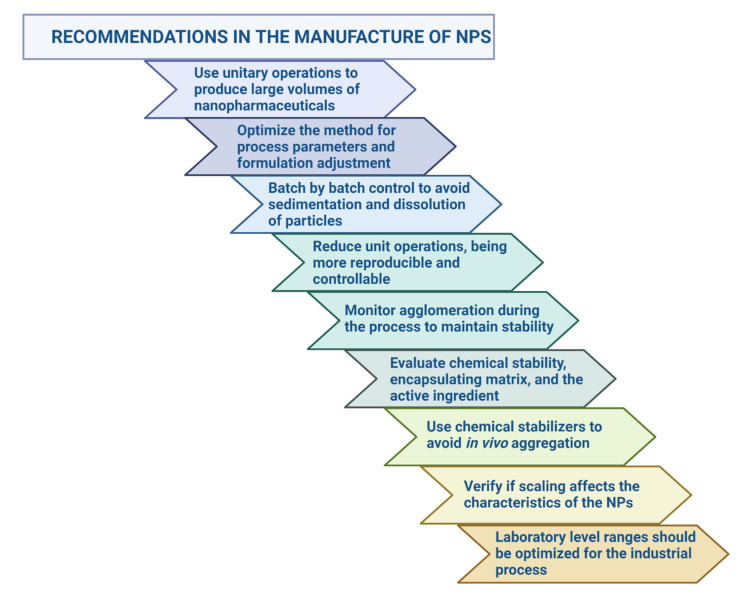
Recommendations for the manufacture of nanoformulations. Created with BioRender.com (accessed on 10 December 2021).

**Figure 11 pharmaceutics-14-00247-f011:**
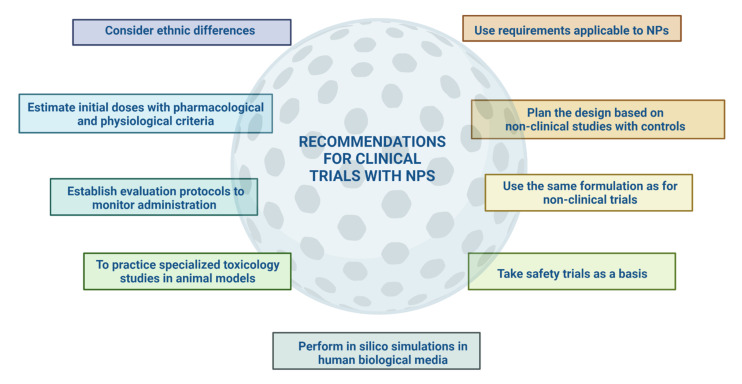
Recommendations for evaluating NPs for clinical trials. Created with BioRender.com (accessed on 10 December 2021).

**Table 1 pharmaceutics-14-00247-t001:** Comparative table with main regulatory agencies and their regulatory progress in nanomedicines.

Aspect	Food and Drug Agency United States of America	European Medicines Agency European Union	Ministry of Health, Labour and Welfare, MHLW
Biopharmaceutical application with innovative technologies with NPs	The Investigational New Drug (IND) application and New Drug Application (NDA) Reformulated Drug Products: Abbreviated New Drug Application (ANDA)	Investigational Medicinal Product Dossier (IMPD)	Investigational New Drug (IND) application or New Drug Application (NDA) under the format: Common Technical Document (CTD)
Nanomedicines Characterization Laboratory	US-NCL	EU-NCL	Research Center for Functional Materials
Guidelines for nanomedicines	Guidance for Industry on Drug Products, Including Biological Products that Contain Nanomaterials–Guidance for Industry Liposome Drug Products Guidance for Industry CDER and CBER Guidance	Reflection Paper on: Nanotechnology-Based Medicinal Products for Human Use Liposomal products Surface coatings Iron-based nano-colloidal products	Guideline for the Development of Liposome Drug Products Reflection paper on the development of block copolymer micelle medicinal products (MHLW/EMA) Reflection paper on silencing ribonucleic acid (siRNA)
References	[47,75,76]	[65,66,67,68,69]	[70,71,72]

Note: IND: The Investigational New Drug; NDA: New Drug Application; ANDA: Abbreviated New Drug Application; US-NCL: Nanotechnology Characterization Laboratory, United States of America; CDER: Center for Drug Evaluation and Research; CBER: Center for Biologics Evaluation and Research; IMPD: Investigational Medicinal Product Dossier; EU-NCL: European Nanomedicine Characterization Laboratory; CTD: Common Technical Document; MHLW: Ministry of Health, Labour and Welfare, Japan; EMA: European Medicines Agency; siRNA: small interfering ribonucleic acid.

**Table 2 pharmaceutics-14-00247-t002:** Standardized tests for the nanotechnological characterization of NPs formulations.

Attribute	Technic	Standard	References
Size/Size distribution	Dynamic Light Scattering (DLS)	ISO 22412: 2017	[133]
		ASTM WK54872 *	[134]
		NCL Joint Assay Protocol, PCC-1	[135]
	Electron Microscopy (TEM, SEM)	ISO 21363: 2020	[136]
		ISO 19749: 2021	[137]
	Atomic Force Microscopy	ASTM E2859-11 (2017)	[138]
	Photon Correlation Spectroscopy (PCS)	ASTM E2490-09 (2021)	[139]
	Electrospray Differential Mobility Analysis (ES-DMA)	NCL Joint Assay Protocol, PCC-10	[140]
	Differential Centrifugal Sedimentation (DCS)	ISO 13318-2: 2007 ***	[141]
		ISO 13318-3: 2004 **	[142]
Size/Size distribution Particle by particle	Nanoparticle tracking analysis (NTA)	ASTM E2834-12	[143]
	Aperture/orifice tube method	ISO 13319-1: 2021	[144]
	Tunable resistive pulse sensing (TRPS)	ISO/CD 13319-2*	[145]
Nanoparticle Composition	Reversed-Phase High Performance Liquid Chromatography (RP-HPLC)	NCL Joint Assay Protocol, PCC-14	[146]
	Mass spectrometry (LC-MS)	ISO/TS 11251: 2019	[147]
Purity	Plasma mass spectrometry	ISO/TS 13278: 2017	[148]
	Chromatography	Scientific publications ****	-
Nanoparticle concentration	Spectradyne nCS1^TM^	NCL Method PCC-20	[149]
Surface Chemistry	Zeta Potential	NCL Method PCC-2	[150]
		ISO/TR 19997: 2018	[151]
	Asymmetric-Flow Field-Flow Fractionation (AF4)	ISO/TS 21362: 2018	[152]
		NCL Method PCC-19	[153]
	Brunauer Emmet Teller	ISO/DIS 9277 *	[154]
Aggregation/agglomeration status	DLS	ISO 22412: 2017	[133]
	Ultraviolet (UV)-visible (vis)	ISO/TS 17466: 2015 **	[155]
	Particle Tracking Analysis	ISO/WD 19430 *	[156]
Free and encapsulated drug	HPLC with UV-vis	ISO/TR 18196: 2016	[157]
	Stable Isotope Tracer Assay		
Shape	Electron Microscopy	ISO 21363: 2020	[136]
		ISO 19749: 2021	[137]
Solubility	pH, Electric conductivity	ISO/TR 13014: 2012/COR 1: 2012 **	[158]

Note: * Under Development; ** Confirmed; *** In Review; **** To our knowledge, there is no standard published by ISO or ASTM. WK: Work Item; PCC: Physico-Chemical Characterization; E: Edition; CD: Committee Draft; TS: Technical Specifications; TR: Technical Report; WD: Working Draft; COR: Technical Corrigendum.

**Table 3 pharmaceutics-14-00247-t003:** Industrial NP production methods.

Approaches	Technique	Uses	Ref.
Bottom-up	Aerosol based processes	Useful for large-scale and multiscale nanoparticle design Aggregation control Size distribution control	[336]
	Atomic or molecular condensation (gas condensation)	Used for metallic NPs It can be combined with chemical vapor deposition (CVD) Allows adjustment of physicochemical properties NPs formation is random	[337]
	Plasma processes	Reduces NPs agglomeration Produces NPs on a large scale Low cost and large volumes Used for NPs in cancer	[338]
	Chemical vapor deposition (CVD)	For inorganic NPs Reactor scale-up is easy Low production cost against large product volumes	[339]
	Liquid phase technique: Sol-gel	Generates solid colloidal particles in a range of less than 100 nm. They are formed from a liquid phase to gel Hydrolysis, condensation and polymerization phases Low-temperature synthesis and flexible design	[340]
Top Down	Grinding and mechanical grinding	Used to produce a nanopowder It is difficult to control contamination and shape NP size can be controlled Easy to scale up	[341]
	Microfluidization	Used for industrial production of liposomes High volume production Uses high pressure and size control Can damage product integrity	[342]
	Electrospray	Produces smaller and more uniform particle size Easy to control parameters Used for bulk production Most used for polymeric nanoparticles	[343]

## Data Availability

Not applicable.

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
