# Peer review of "The Hitchhiker’s Guide to Human Therapeutic Nanoparticle Development"

_pharmaceutics, 2022, doi:10.3390/pharmaceutics14020247_

Round 1

Reviewer 1 Report

This should be a comprehensive guideline how to make and assess the stability, toxicity, functionality, etc of  therapeutic nanoparticles, but I find it hard to read with a lot of explanations and not explaining nothing. so each statement should have examples in brackets of one word, word, word, so the readers would be interested in reading and exploring further, all the shortnames should be explained, figures and tables are mentioned close to the reference in the sentence so the reader cannot understand is the table cited from that reference or from the text, remove dots from the figures and tables, check one reference formatted differently than the other, try to shorten the text, like researchers have demonstrated, bla bla, in the statement, that caco2 cells treated with... nanoparticles in concentrations... proved or demonstrated something.

Author Response

Examples in brackets added:

The toxicity of NPs is related to evaluating their physical and chemical characteristics and their relationship with adverse events such as thrombosis and platelet aggregation (C60CS, SWNT and MWNT carbon NPs used in drug discovery and delivery affect vascular hemostasis and precipitate causing thrombosis).

The main challenge in calculating EE is the accuracy of the drug analysis.  There are complications in the use of both direct and indirect procedures, and a combination of both types is preferable to obtain more accurate results (drug-loaded solid lipid NPs were separated by centrifugation before measuring the unencapsulated drug in an aqueous phase, in other studies, the drug content was directly measured to determine EE%).

The physiological environment of the administration site (pH or temperature) can accelerate or decelerate the release of the encapsulated molecule, and it is crucial to consider the environment to which the NPs will be exposed in an in vivo model (protein‑based hydrogel biocomposites were pH-sensitive, and their degree of swelling was significant at pH 7.4 and a temperature of 37 °C)

PDF comments:

  1. Change in the title

The Hitchhiker’s guide for human therapeutic nano-particles development

  1. This is only FDA guideline, where are the others?

References 55-62 were added and placed as follows:

Different agencies, such as the EMA [55-59] and the Japanese Ministry of Health, Labor and Welfare (MHLW)/ Pharmaceuticals and Medical Devices Agency (PMDA) [60-62], have published several guidelines for nanotechnology-based medical products over the past few years, including Reflection Papers for liposomes, micelles, and nanoparticles

  1. When the Table 1 is here in brackets near the reference 44, it seems that we need to look at that reference for table 1

Reference 44 was replaced by references 55-59 and 60-62, respectively.  The reference to table 1 was placed at the end of the idea.

Different agencies, such as the EMA and the Japanese Ministry of Health, Labor and Welfare (MHLW)/ Pharmaceuticals and Medical Devices Agency (PMDA), have published several guidelines for nanotechnology-based medical products over the past few years, including Reflection Papers for liposomes, micelles, and nanoparticles (Table 1).

  1. Cite 53 without format

Reference 63, previously referenced as 53, has been updated.

  1. Table can be more comprehensive so the text is easier to read, for each agency, there should be reference to check what they are recommending

Table 1 was oriented vertically to make it wider, and a row of references was added for each agency.

  1. Explain, what kind of demonstrations, what kind of aspects?

The demonstration of biological activity in in vitro assays, i.e. cell lines, should correlate with the in vivo assay and the selection of the tissue and animal model, considering the first aspects of the therapeutic product.

  1. And you will now explain those systems and challenges?

In the following, we describe each of the parameters and their challenges in obtaining NPs for biomedical applications.

  1. Toxicity comes after particle analysis

The idea was clarified to explain that the toxicity assessment of NPs starts with the physicochemical characterization:

Unlike the development of other therapeutic products, the evaluation of the toxicity potential of NPs in biological systems begins with the complete physicochemical characterization, being a critical step in the early stages of development, which contributes to the principles of quality, safety, and efficacy [63,66].  To make a product effective in the clinical setting, it is necessary to employ appropriate characterization technologies that correlate effect and biological consequences and predict toxicologic and therapeutic outcomes at the early stage of product development…

…Inflammation, neurodegenerative or circulatory disorders, and other disorders are further adverse events described.

  1. Again the same, it is the figure from this manuscript and not from the reference 56

Figure 1 and its reference were relocated.  Figure 1 is now referenced as follows:

The main reason why their approval is limited for use in human therapeutics is due to incomplete particle analysis.  The characterization should include determining size, shape, composition, charge and surface chemistry, encapsulation efficiency, and evaluating the encapsulated drug (Figure 1) with parameters such as loading, distribution, re-lease kinetics, interaction with cells, and transport system.

  1. “It has been described, for example, for gels the influence PSD and zeta potential on size”. Strange long sentence

For gel nanoformulations (chitosan gel containing acrylic-based nanocapsules), it has been demonstrated that the zeta potential of the NPs influence the viscosity, rigidity, and gel network structure.  In the case of tablets (dexamethasone-loaded PLGA NPs embedded in alginate), the encapsulation efficiency, morphology, diameter of the particles can affect the thickness of the formulation.

  1. Figure 1. “When evaluating these attributes is crucial to”.  Strange part of the sentence

The following sentence was relocated as part of the explanation for figure 2 (see comment 12).

To evaluate these attributes, it is essential to use the most appropriate combination of techniques in each case.

  1. What about figure 2? why you did not put the figure 2 where you are mentioning the figure 2?  It should be written in the figure 2 whose recommendations are those explained in the figure

Before the regulatory aspects of physicochemical characterization (section 3.2), the following paragraph and figure 2 were added.

The complete characterization of NPs considers particle-related parameters (size distribution, morphology, zeta potential, composition, charge, and surface chemistry) and entrapment parameters (drug charge and distribution, encapsulation efficiency, release kinetics and cell interaction).  To evaluate these attributes, it is essential to use the most appropriate combination of techniques in each case (Figure 2).

In addition, references to the recommendation figures in each section have been added to the text:

  • In vitro We can summarize that the specific evaluation of these systems to determine the in vitro toxicity of formulations presents several difficulties for in vitro assays as there are many differences in the in vitro and in vivo study designs (Figure 5).
  • In vivo The safety and therapeutic efficacy of nanoparticles can only be assessed by rigorous in vivo testing and based partly on the results obtained from physicochemical characterization studies and in vitro assays (Figure 6).
  • With increasing technology and experience gained, the industry has production methods aimed at reducing the unit operations in the synthesis of NPs (Figure 10), being more reproducible and controllable, and reducing the obstacles in the manufacture of nanomedicines.
  • Clinical trials. Regulatory authorities suggest that the sponsor evaluates any changes in the manufacturing process, or at any stage of its clinical development, related to the drug product or its formulation to determine whether the changes impact the product’s safety (Figure 11).
  1. References where the protocols can be found?

A column of references was added to table 2.

  1. Dots are not necessary

In Table 2, the dots at the end of each idea were removed.

  1. It is the figure 3 from this manuscript and not from the reference 135

Reference 163 (formerly 135) was removed because it was repeated.  Thus, the reference to figure 3 is as follows:

The composition of this corona influences the biological activity of the encapsulated molecule and its release kinetics [164], as depending on the concentration and type of proteins present, the circulation time may increase or decrease (Figure 3).

  1. Figure 10. Dots are not needed

Dots at the end of each idea were eliminated.

  1. Reference 160 formatted differently

The format of the reference was corrected.

Further changes:

  1. Figures 6 and 7 were reordered
  2. Table 3 was relocated
  3. References to figures 9, 10 and 11 were relocated, not close to the references.

Reviewer 2 Report

In this manuscript, the authors presented a comprehensive review on developing NP systems for human therapeutic applications. The synthesis, characterizations, regulation of the functions, and in vitro as well as in vivo applications of NP systems were introduced and discussed in great detail. It is a very interesting topic. This work will be very helpful for readers in the fields of nanotechnology, biomedicine, bioanalysis, tissue engineering, and others. This manuscript is recommended for publication after a few questions are addressed.

Special comments for the revision:

  1. It is suggested for the authors to add more introduction and discussion on previous reviews on using NP systems for therapeutic applications, and then indicate clearly the novelty and significance of this manuscript.
  2. What are the advantages and disadvantages of therapeutic NP systems compared to microscale particle systems? More details should be added.
  3. The authors are suggested to provide more introduction on the types of NP systems that could be utilized for therapeutic applications.
  4. More information on the functionalization of NP systems for improved therapeutic applications could be discussed further.

Author Response

Special comments for the revision:

  1. It is suggested for the authors to add more introduction and discussion on previous reviews on using NP systems for therapeutic applications, and then indicate clearly the novelty and significance of this manuscript.

There are currently 58 nanoparticle therapies and imaging agents approved by major regulatory agencies for clinical use.  These formulations offer promising results for treating a wide variety of diseases such as cancer, infections, autoimmune disorders, cardiovascular, pulmonary, neurodegenerative, ocular (glaucoma) and regenerative therapy, among other applications.  The most successful formulations to date have been polymers and lipids.  The main advantages of nanoparticles are (1) increased bioavailability due to improved water solubility, (2) increased resistance time in the body (increased half-life for clearance/increased specificity for its cognate receptors), and (3) targeting of the drug to a specific region of the body (its site of action).

Of the drugs approved by the major regulatory agencies, the FDA and EMA all use liposomal nanoparticle systems except Abraxane, an albumin-bound paclitaxel nanoparticle.  In the case of proliferative or cancerous conditions, many approved formulations apply to various stages of the disease.  Among these drugs, the following stand out: Doxil, liposomal doxorubicin functionalized with polyethylene glycol (PEG), liposomal daunorubicin (DaunoXome), liposomal vincristine (Marqibo) and liposomal irinotecan (Onivyde), liposomal doxorubicin without PEG (Myocet) and liposomal mifamurtide (MEPACT).  Most of these formulations are non-PEGylated, despite the recognized advantages that this encapsulation system offers.  Marqibo® (Vincristine sulfate), approved by the FDA to treat acute lymphocytic leukemia in adults, is a liposomal formulation based on sphingomyelin and cholesterol, which greatly improves circulation time accelerates dose escalation compared to standard Vincristine.  Kadcyla® (Herceptin®) is an antibody-drug conjugate for treating HER2+ breast cancer.  The drug is delivered to cancer cells by recognizing the HER receptor (transtuzumab), and maytansine (DM1) triggers apoptosis.

Polymeric, non-polymeric nanoformulations and liposomes have also been developed for infectious diseases.  Example of these already approved drugs Lipoquin™, the liposomal formulation of ciprofloxacin inhaled formulation and release up to 24 hours.  Ambisome® (amphotericin B) and others for lipid-containing amphotericin B, such as Abelcet, Visudyne.  Recent examples of Covid-19 vaccines packaged in liposomal systems are the Pfizer/BioNTech and Moderna COVID-19 vaccines, both of which are formed in liposomal nanoparticles (LNP) or PEGLips (artificial phospholipid vesicles effective in stabilizing pharmaceutical products).  These vaccines make it possible to stabilize mRNA thanks to their lability.

In iron replacement therapeutics, nanoparticles have also had a significant impact on increasing iron concentrations in the body and are considered complex non-biological drugs.  Also, nanoparticle systems in autoimmune conditions are promising as they target inflamed tissue.  An example is Certolizumab pegol (CZP), a TNF-α inhibitor widely used in the clinic with a half-life of 14 days.

We can conclude that in terms of therapeutic applications for nanoparticles, in 2016, the number of approved nanoparticles used in the clinic was 51 nanomedicines and in another publication in November 2021, it appears updated to 58.

The novelty and importance of the manuscript are referred to in the following paragraphs of the introduction and conclusions:

Introduction

This manuscript describes the major challenges researchers have faced when creating a new formulation.  Our work aims to provide helpful information to improve the success of nanomedicines by compiling the challenges described in the literature that support the development of this novel encapsulation system.  Additional guidance is required to cover the particularities of this type of product, as some challenges in the regulatory framework do not allow an accurate assessment of NPs with sufficient evidence for clinical success.  We also reflect on the current regulatory standards required to approve these biopharmaceuticals and the requirements demanded by regulatory agencies.  We propose a step-by-step approach to the different stages of nanoformulation development, from initial design to the clinical stage, exemplifying the distinct challenges and the measures taken by regulatory agencies to respond to these challenges.

Conclusions

For that reason, with this manuscript, we wish to unify the different challenges described in the formulation of human-targeted nanoparticles and regulatory regulations.  The document describes each challenge separating each of the stages of development of nanomedicines and complements each section with the regulations, protocols, guidelines, guidelines, regulatory standards described to date for each section.  We have not been able to locate in the literature a similar document that allows us to orientate researchers who develop this type of biopharmaceutical completely.  Although some authors have addressed the subject, it is not yet unified in a single text, with the difficulties or the regulatory scope appearing in separate documents.

  1. What are the advantages and disadvantages of therapeutic NP systems compared to microscale particle systems?  More details should be added.

Micrometer and nanometer-scale encapsulations are transport systems that create a physical barrier to protect the active ingredient from the external environment.  Microparticle formulations increase the bioavailability of the drug but have several drawbacks: low encapsulation efficiency, abrupt or incomplete release of the active ingredient, reduced biological activity, among others.  Nanoencapsulation has been considered a more efficient transport system than microscale systems, with better functionality because it exhibits greater drug protection, increased stability, higher loading capacity, superior encapsulation efficiency, sustained release, and improved bioavailability.  The limiting aspects of therapeutic use for nanoformulations are the control of drug release, particle opsonization, and the toxicity and immunogenicity they may cause.  In studying the relationship between PLA-PEG particle size and its transport efficiency across the nasal mucosa, tetanus toxoid was encapsulated in particles of different sizes (200 nm, 1.5 μm, 5 μm and 10 μm).  The nasal bioavailability of tetanus toxoid encapsulated in 200 nm nanoparticles was higher than in larger particles.

  1. The authors are suggested to provide more introduction on the types of NP systems that could be utilized for therapeutic applications.

Dendrimers are branched polymers with unique topological and structural characteristics, as they have three parts: a focal core, building blocks with several inner layers with repeating units, and multiple peripheral functional groups.  Liposomes are spherical structures consisting of one or more lipid bilayers enclosing aqueous spaces.  Polymeric micelles are nanostructures formed by the spontaneous arrangement in an aqueous medium of amphipathic polymer macromolecules.  Encapsulation in NPs consists of trapping active ingredients using a surrounding material.  This technology allows obtaining nanospheres (deposition systems that incorporate the active ingredient in the particle-matrix) or nanocapsules (matrices formed by the drug as the nucleus, and the particle material as the capsule covering).

  1. More information on the functionalization of NP systems for improved therapeutic applications could be discussed further.

Functionalization consists of the conjugation of molecules to the surface of the particles, such as folic acid, biotin, oligonucleotides, peptides, monoclonal antibodies, functional groups, among others.  This modification makes it possible to incorporate or alter specific properties in the NPs with great precision.  The functionalized particles possess non-invasive, anti-agglomeration characteristics and good physical properties.  The bonding can be done through covalent and noncovalent bonds.  With noncovalent conjugation, it is possible to make changes without affecting the structure of the molecules.  For example, Yue et al. (2019) developed a noncovalent functionalization process of curcubit[7]uril to gold NPs, enabling image-driven chemo-photothermal therapy.  In the case of covalent bonding, structures with multiple functions aimed at diagnostic and theranostic therapy can be obtained and use linker molecules such as polyethylene glycol (PEG).  An example is a work of Chen and co-workers (2017), who synthesized micelles conjugated with antigen-binding (Fab) fragments of antibodies, using PEG as a spacer to obtain a high Fab density on the surface of the NPs and achieved higher bioactivity.

By functionalization, it is possible to modify properties such as surface chemistry, hydrophobicity or charge of the NPs to improve their solubility, biocompatibility, biodistribution and clearance.  The two difficulties of NPs that have been mostly worked on with functionalization are uptake and biocompatibility; the efficiency of their uptake, the cytotoxic effects of the particles on cells, and their ability to cross biological barriers limit the clinical use of these systems.  The cellular uptake (active or passive) of NPs depends on their physicochemical properties, so it is possible to increase the stability and reduce the aggregation of the particles through functionalizations with PEGs, peptides, or zwitterionic ligands, to increase passive uptake.  To achieve active and targeted cellular uptake, it can also be conjugated with antibodies, aptamers, carbohydrates, and proteins [14].  Fathian kolahkaj et al. (2019) reported effective uptake of PLGA NPs modified with monoclonal antibodies against Human epidermal growth factor receptor 2 (HER2), with increased levels of internalization in HER2-positive cell lines.  Other ligands, such as transferrin, insulin and lipoproteins, can bind to the surface of the NPs to cross biological barriers such as the blood-brain barrier.  Lactoferrin functionalization of trimethylated PLGA-chitosan NPs encapsulating huperzine A had increased cellular uptake, with improved drug transport to the brain.   In the case of biocompatibility, conjugation of molecules on the surface of NPs has shown improvement in formulations by modifying the surface charge and inactivation of chemical groups that destabilize the cell membrane.  For example, the addition of molecules such as human albumin allows reducing the toxicity of the particles.  Sanità et al. in 2020 reported that functionalization with albumin in hybrid NPs of silver and eumelanin suppressed in vitro hemotoxicity in normal mammary cells (MCF10a).

Reviewer 3 Report

The review entitled “The Hitchikers guide to developing human therapeutic nano-particles” discusses the therapeutic effect of nanoparticles. The approach they have and the proposal on how to proceed in the development of new formulations is very interesting.

Regarding the animal model, the authors report which are the most important for invertebrates and vertebrates. Some studies report that the pig is also a good model, better than the mouse at a phylogenetic level. Could the authors say anything about it?

Author Response

Regarding the animal model, the authors report which are the most important for invertebrates and vertebrates.  Some studies report that the pig is also a good model, better than the mouse at a phylogenetic level.  Could the authors say anything about it?

Another animal model (higher organism) used with advantages in in vivo assays with NPs is the pig (domestic Sus scrofa).  This mammal shortens the phylogenetic distance between the rodent and human models due to its similarity with the immune and lymphatic systems.  Acute hypersensitivity reactions (HSRs) induced by intravenous (IV) drugs and other compounds represent an ancient, unresolved immune barrier.  The swine model has been proposed by regulatory agencies for preclinical risk assessment of HSRs in the clinical stages of nanodrug development as predictors of adverse drug reactions (ADRs) and severe adverse events (SAEs).  The porcine model of complement activation-related pseudoallergy (CARPA) is a classical one, which determines the immune reactivity of nanomedicines.  It has also been used in safe infusion protocols for reactogenic Nps such as liposomal drugs (PEGylated liposomal prednisolone (PLP), which can provoke HSRs, with an exacerbated and toxic response. Another example is the administration of solid lipid NPs encapsulating nucleic acid, ONPATTRO® (Patisiran), approved by the FDA and EMA.  However, as rodents differ in their metabolism and physiology from humans, swine faithfully reproduces the human organism like no other animal model.  Animal models should be selected considering aspects such as correspondence with the route of administration, dose, experimental design, physiological state, and the stability of the nanomaterial in biological media.